# ATP released by intestinal bacteria limits the generation of protective IgA against enteropathogens

Michele Proietti[1,8], Lisa Perruzza[1], Daniela Scribano [2,3], Giovanni Pellegrini[4], Rocco D'Antuono[1], Francesco Strati [1], Marco Raffaelli[1], Santiago F. Gonzalez[1], Marcus Thelen[1], Wolf-Dietrich Hardt[5], Emma Slack[5], Mauro Nicoletti[2] & Fabio Grassi [1,6,7]

T cell dependent secretory IgA (SIgA) generated in the Peyer's patches (PPs) of the small intestine shapes a broadly diverse microbiota that is crucial for host physiology. The mutualistic co-evolution of host and microbes led to the relative tolerance of host's immune system towards commensal microorganisms. The ATP-gated ionotropic P2X7 receptor limits T follicular helper (Tfh) cells expansion and germinal center (GC) reaction in the PPs. Here we show that transient depletion of intestinal ATP can dramatically improve high-affinity IgA response against both live and inactivated oral vaccines. Ectopic expression of *Shigella flexneri* periplasmic ATP-diphosphohydrolase (apyrase) abolishes ATP release by bacteria and improves the specific IgA response against live oral vaccines. Antibody responses primed in the absence of intestinal extracellular ATP (eATP) also provide superior protection from enteropathogenic infection. Thus, modulation of eATP in the small intestine can affect high-affinity IgA response against gut colonizing bacteria.

[1] Institute for Research in Biomedicine, Università della Svizzera Italiana, Via Vincenzo Vela 6, 6500 Bellinzona, Switzerland. [2] Department of Medical and Oral Sciences and Biotechnologies, University "Gabriele D'Annunzio", Via dei Vestini, Campus Universitario, 66100 Chieti, Italy. [3] Department of Public Health and Infectious Diseases, University "La Sapienza" of Rome, Piazzale Aldo Moro 5, 00185 Rome, Italy. [4] Laboratory for Animal Model Pathology, Institute of Veterinary Pathology, Vetsuisse Faculty, University of Zurich, Winterthurerstrasse 268, 8057 Zurich, Switzerland. [5] Institute of Microbiology, ETH Zürich, Vladimir-Prelog-Weg 4, 8093 Zürich, Switzerland. [6] Istituto Nazionale Genetica Molecolare "Romeo ed Enrica Invernizzi", Via Francesco Sforza 35, 20122 Milan, Italy. [7] Department of Medical Biotechnology and Translational Medicine (BIOMETRA), Università degli Studi di Milano, Via Vanvitelli 32, 20129 Milan, Italy. [8] Present address: CCI-Center for Chronic Immunodeficiency, Universitätsklinikum Freiburg, 79106 Freiburg, Germany. These authors contributed equally: Michele Proietti, Lisa Perruzza. Correspondence and requests for materials should be addressed to F.G. (email: fabio.grassi@irb.usi.ch)

nteric pathogens such as enteropathogenic *E. coli* and non-typhoidal *Salmonella* are a major health burden in both humans and animals. The rapid spread of antibiotic resistance in these species highlights the need for better disease prophylaxis. Protection from infection is most effective when strong mucosal immune responses have been induced, either by prior infection or by oral vaccination[1,2]. High-affinity secretory IgA (SIgA) promotes enteropathogen enchainment and aggregation to disable and clear potentially invasive species from the intestinal lumen[3]. However, balancing safety of the vaccination strain with sufficient immune stimulation has proved challenging[4].

T follicular helper (Tfh) cells express high levels of the ATP-gated P2X7 receptor, a non-selective cationic channel that opens to form a cytolytic pore when exposed to micromolar concentrations of extracellular ATP (eATP). P2X7 activity therefore controls Tfh cells abundance in Peyer's patches (PPs): Resistance of Tfh cells to ATP-mediated cell death by deletion of P2X7 enhances germinal center (GC) reactions[5]. As eATP is produced in large quantities by the intestinal microbiota, this directly suppresses commensal-specific IgA responses primed in the gut-draining lymphoid tissues and affects microbiota composition[6]. This study is based on the hypothesis that similar effects may dampen immunity against enteric pathogens and oral vaccines.

We show that ATP released by intestinal bacteria permeates the intestinal epithelium and can be found at high concentrations in hepatic portal blood. Eliminating this eATP, via administration of apyrase, dramatically improves the induction of specific IgA in response to either *Salmonella* infection or an inactivated oral vaccine. We could not measure any adverse effects of altered anti-microbiota immunity secondary to oral apyrase administration, suggesting that apyrase application is safe. Moreover, these enhanced immune responses provide superior protection from secondary infection.

## Results

### ATP released by microbiota affects Tfh cells in PPs via P2X7.

In the small intestine and portal vein of specific pathogen free (SPF) mice, we measured micromolar concentrations of eATP that was detected at much lower levels in germ-free (GF) mice or in other circulatory districts (Fig. 1a, b). To address the contribution of the epithelium to eATP in the small intestine, we induced epithelial regeneration in the ileum by starvation and re-feeding, as described[7]. In the presence of bacteria, the variations in epithelial turnover by starvation and re-feeding corresponded to undistinguishable concentrations of ileal eATP. In the absence of bacteria, starvation did not affect the percentage of proliferating cells[8]. However, the concentration of ileal ATP was dramatically reduced with respect to SPF mice with comparable amount of proliferating epithelial cells, suggesting that the great majority of eATP measured in the ileal lumen is of bacterial origin (Supplementary Figure 1a, b). Therefore the microbiota generates high levels of eATP that can penetrate into the intestinal epithelium and draining blood. We cannot exclude that fungi, archaea, and protozoa might also contribute to the eATP present in the intestinal lumen. Consistent with other reports[9,10]; eATP was detectable in cultures from different bacterial strains isolated from *ilea* of our mouse colony (Fig. 1c) and could be acutely exacerbated by vancomycin/ampicillin/metronidazole (VAM) treatment (Fig. 1d, e). In vivo VAM administration resulted in an acute significant increase of eATP in the ileum and portal vein blood (Fig. 1f). In wild type (WT), but not *P2rx7*[−/−] mice, VAM administration-induced enhanced phosphatidyl-serine (PS) exposure in Tfh cells from PPs (Fig. 1g and Supplementary Figure 1c, d), suggesting bacteria-derived eATP can modulate high-affinity SIgA response. Antibiotic treatment can

be contraindicated in acute bacterial gastrointestinal infections due to negative effects on microbiota recovery. This data further suggest that antibiotic treatment may negatively effect the induction of T-cell dependent intestinal immunity in these infections.

### Enhanced SIgA response by depletion of eATP.

The IgA response to *E. coli* is dependent on Tfh cells in PPs[11] and is significantly enhanced in *P2rx7*[−/−] mice[5], suggesting that P2X7 activity can affect the T-cell dependent SIgA response. To address whether depleting bacteria-derived ATP could influence T cell-dependent IgA responses via P2X7, we used a recombinant *E. coli* strain (*E. coli*[pApyr]) carrying an expression plasmid for *Shigella flexneri's* periplasmic ATP-diphosphohydrolase (apyrase) (Supplementary Figure 2a–d)[12,13]. The supernatant of *E. coli*[pApyr] cultures showed ATP-degrading activity that was absent in *E. coli*[pBAD28] (Fig. 2b); its fractionation resulted in the recovery of apyrase activity within outer membrane vesicles (OMVs), suggesting the enzyme was released in the extracellular space (Fig. 2a, c). Notably, eATP[14,15] was undetectable in cultures of *E. coli*[pApyr], indicating that apyrase efficiently degraded ATP (Fig. 2d). To address whether bacteria-derived ATP could selectively limit SIgA responses in the small intestine of normally colonized animals, we administered *E. coli*[pApyr] or *E. coli*[pBAD28] as control, to SPF mice by orogastric gavage (Supplementary Figure 3b). Administration of *E. coli*[pApyr] resulted in the increase of Tfh cells in the PPs concomitant to reduced Annexin V staining in flow cytometry, suggesting degradation of bacterial ATP reduced Tfh cell death via P2X7 receptor (Fig. 3a). As expected, we observed a poor SIgA response to *E. coli* in mice gavaged with *E. coli*[pBAD28][16]. However, anti-*E. coli* IgA was significantly increased in mice gavaged with *E. coli*[pApyr] (Fig. 3b, c), despite identical intestinal *E. coli* load (Supplementary Figure 3a), indicating that abrogation of ATP release by bacteria results in the development of high-affinity IgA responses. The analysis of IgA in intestinal washes from mice gavaged with the two *E. coli* transformants on different bacterial species revealed lack of detectable reactivity (Supplementary Figure 4a). Moreover, intestinal IgA from untreated and immunized mice stained an analogous percentage of commensals from WT mice (Supplementary Figure 4b), indicating the absence of epitope-spreading to resident microbiota members.

To further address the role of ATP released by bacteria in modulating the SIgA response, we monitored endoluminal ATP after orogastric administration of *E. coli*[pBAD28] and *E. coli*[pApyr] in mice maintained with Chloramphenicol and Ampicillin (CA) (a bactericidal mix active on endogenous flora but not on CA-resistant *E. coli*[pBAD28] and *E. coli*[pApyr]) or Penicillin/Streptomycin/Vancomycin (PSV) (bactericidal on both endogenous flora as well as *E. coli* transformants) in drinking water (Fig. 3d). Oral gavaging with *E. coli*[pBAD28] in mice maintained in PSV as compared to CA resulted in a significant acute increase of endoluminal ATP because of bacterial lysis (Fig. 3e). Notably, the analysis of anti-*E. coli* IgA after multiple gavaging in this setting showed that the increase in eATP concomitant to *E. coli*[pBAD28] gavaging in the presence of PSV correlated with reduced anti-*E. coli* IgA with respect to the group treated with non-bactericidal CA (Fig. 3f). In contrast, in mice colonized with *E. coli*[pApyr], ATP degradation by apyrase in both treatment groups (Fig. 3e) resulted in undistinguishable anti-*E.coli* IgA response (Fig. 3f). These data further show that an increased release of ATP by bacteria corresponds to a reduced generation of specific IgA.

### Enhancement of specific SIgA by *Salmonella* vaccine with apyrase.

In the streptomycin mouse model of non-typhoidal

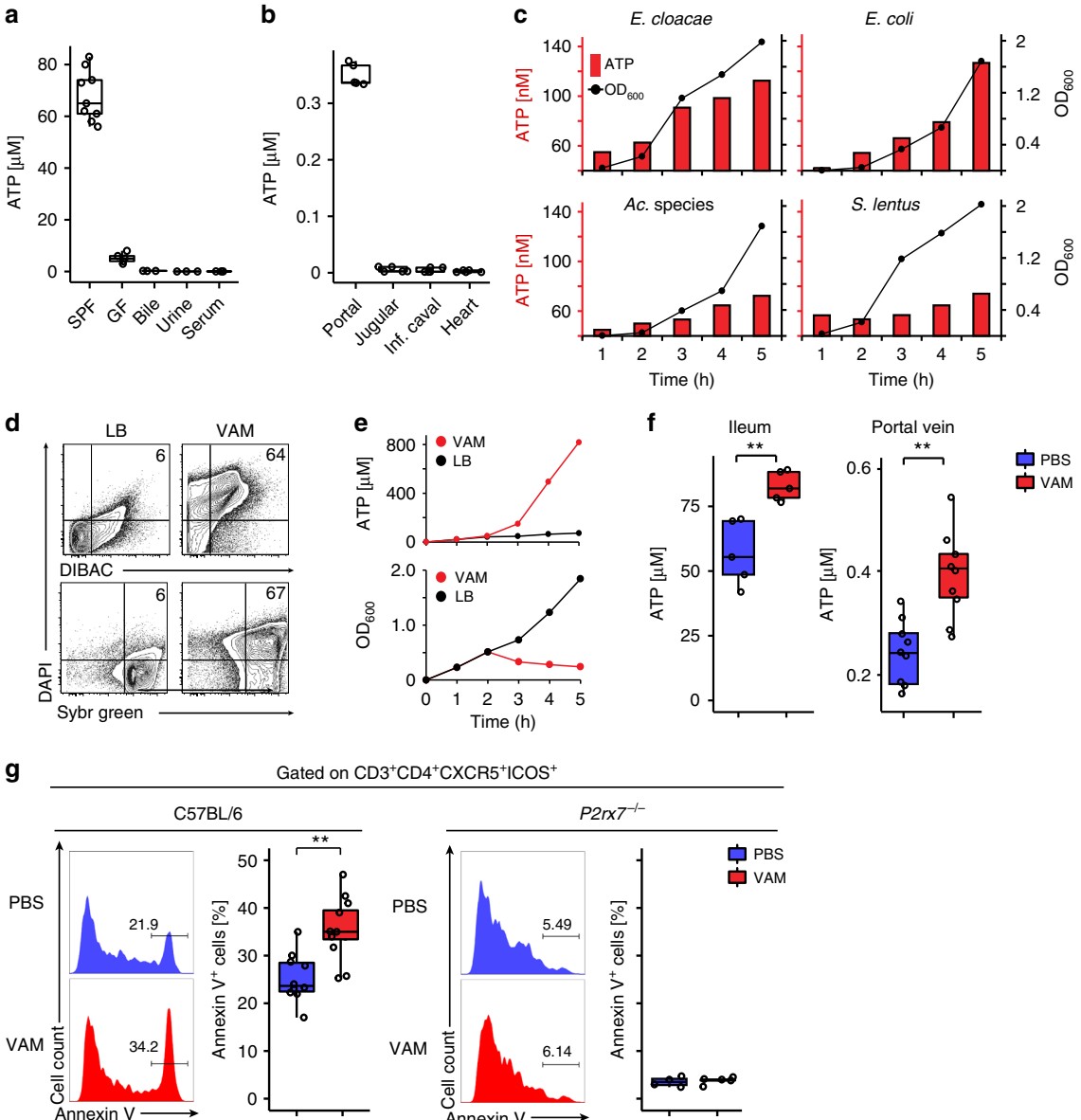

**Fig. 1** Bacterial origin of intestinal ATP. **a** ATP concentration in the lumen of ileum from SPF and GF mice, bile, urine, and serum from SPF mice. **b** ATP concentration in serum from portal, jugular, inferior caval veins, and heart. **c** ATP concentrations in culture medium (bars) and cell growth ($OD_{600}$) of the indicated bacterial species isolated from the small intestine of SPF mice. **d** Flow cytometry of ileal bacteria either maintained in culture medium (LB) or treated with VAM, for membrane damage (DIBAC+DAPI+ cells, upper dot plots) and cell death (SybrGren+DAPI+ cells, lower dot plots). **e** ATP concentrations (upper panel) and ileal bacteria growth (lower panel) in untreated (LB) and VAM-treated cultures. **f** ATP concentration in ileum and portal vein of SPF mice at 3 h after orogastric gavage with PBS or VAM. **g** Representative histograms and statistical analysis of Annexin V+ cells within Tfh cells from PPs of WT (C57BL/6) and $P2rx7^{-/-}$ mice at 3 h after gavage with PBS or VAM. The boxplots show median and upper and lower quartiles. The extreme lines show the highest and lowest value. The boxplot is overlaid with the visualization of single observations. Two-tailed Mann–Whitney $U$-tests. $^{**}p < 0.01$. One representative experiment out of at least three is shown

salmonellosis[17], oral infection with *Salmonella enterica* serovar Typhimurium (*S*.Tm) leads to GALT colonization and systemic dissemination of bacteria, as originally shown with *S. enteriditis*[18]. To address whether apyrase expression in live-attenuated *S*.Tm could increase the specific SIgA response and confer enhanced protection from infection by a virulent strain, we generated an attenuated *S*.Tm strain (ATCC 53648) carrying either pBAD28 (*S*.Tm^pBAD28) or apyrase-bearing pHND10 (*S*.Tm^pApyr) (Supplementary Table 1). As observed with *E. coli*^pApyr, ATP was undetectable in culture medium of apyrase-expressing *S*.Tm^pApyr (Fig. 4a). Notably, Tfh and GC B cells as well as plasma cells secreting IgA specific for *Salmonella* LPS were all significantly

increased in mice immunized with *S*.Tm^pApyr (Fig. 4b, c and Supplementary Figure 5b-d), as was the concentration of anti-*Salmonella* IgA in intestinal wash (Fig. 4d).

**Effective protection from *Salmonella* infection by *S*.Tm^pApyr.** SIgA protects the host from invasion by *S*.Tm or other enteropathogens by limiting the interaction of bacteria with the gut epithelium[19,20]. In the non-typhoidal salmonellosis model, both enchained growth and classical agglutination, requiring high-affinity IgA to cross-link dividing and colliding bacteria, are the main protective effects[3]. As only non-clumped bacteria can

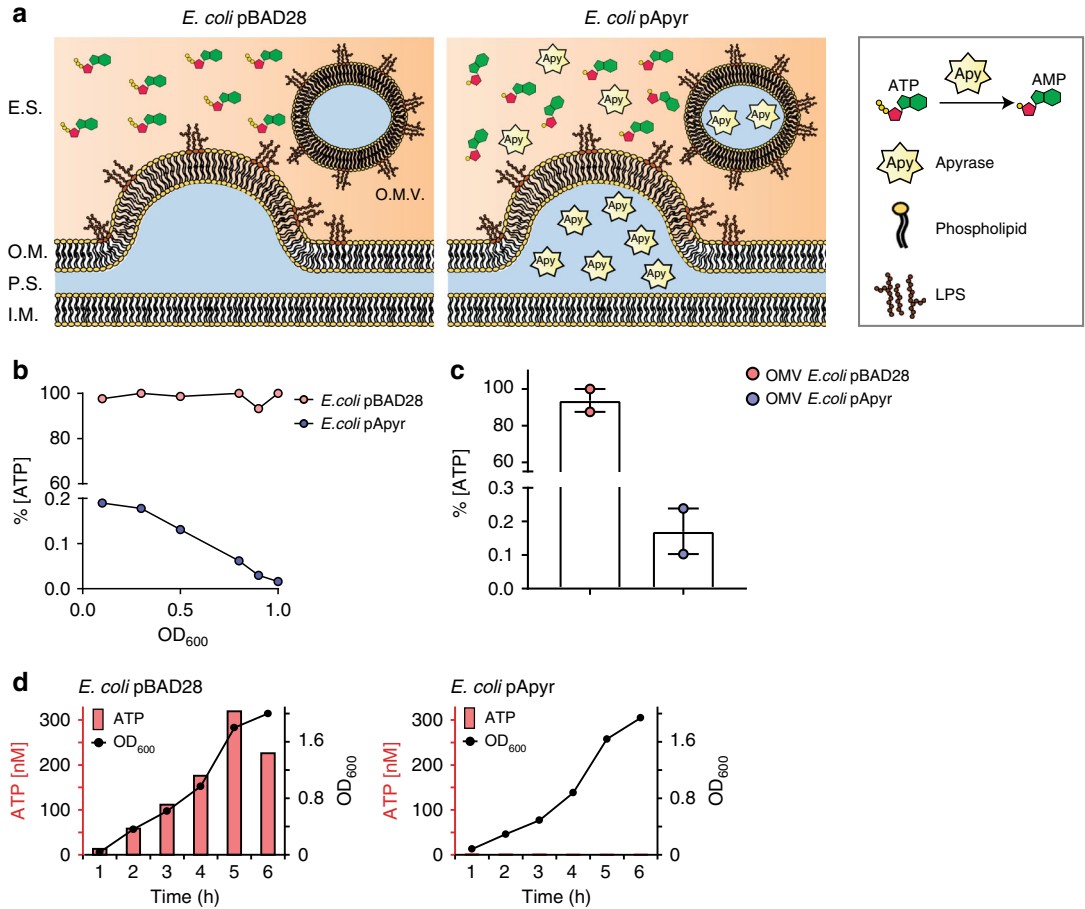

**Fig. 2** Apyrase release by *E. coli*. **a** Schematic model of apyrase release by *E. coli* (E.S., extracellular space; O.M., outer membrane; P.S., periplasmic space; I. M., inner membrane; O.M.V., outer membrane vesicle). **b** Apyrase activity in culture supernatants of the indicated *E. coli* transformants at different $OD_{600}$ expressed as percentage of non-degraded ATP. **c** Apyrase activity in OMVs isolated from *E. coli*[pBAD28] and *E. coli*[pApyr]. **d** ATP concentrations in culture medium (bars) and bacterial growth ($OD_{600}$) over time for *E. coli*[pBAD28] (left panel) or *E. coli*[pApyr] (right panel). L-arabinose was added at time 0. One representative experiment out of at least three is shown

approach the intestinal epithelium and invade into host tissues, this has the effect of hugely reducing the infectious burden in the intestine. We first addressed whether the enhanced IgA responses observed after vaccination with *S*.Tm[pApyr] corresponded with enhanced aggregation of *Salmonella* in the gut lumen. Vaccinated mice were therefore orally infected with 1:1 mixture of GFP-tagged and mCherry-tagged attenuated *S*.Tm (*S*.Tm[att], used to avoid confounding inflammation in the control animals)[21] and cecal content was analysed at 5 h post infection by confocal microscopy. In mice immunized with *S*.Tm[pApyr], we observed a significant decrease in planktonic *S*.Tm[att] as compared to mice immunized with *S*.Tm[pBAD28] (Fig. 4e), and fewer *S*. Typhimurium swimming in the cecal crypts (Fig. 4f).

Correspondingly, when challenge infections were carried out with fully virulent *S*. Typhimurium (*S*.Tm[WT]) in *S*.Tm[pApyr] vaccinated mice, disease parameters, including histopathological score, fecal Lipocalin 2 (LCN2) and GALT infection, were all significantly decreased as compared to controls or *S*.Tm[pBAD28]-vaccinated mice (Fig. 5a–c). *Salmonella* colonizing systemic compartments (e.g. spleen and liver) follows a GALT-independent route of infection (presumably via blood circulation)[22]. Accordingly, infection with *S*.Tm[WT] determines increased permeability of the gut-vascular barrier that is reflected by translocation of FITC-dextran from the intestinal lumen into the bloodstream and liver[23]. Vaccinated mice were significantly more resistant to blood absorption of FITC-dextran administered

via orogastric route with respect to non-vaccinated animals and immunization with *S*.Tm[pApyr] conferred enhanced protection (Fig. 5d). Moreover, *S*.Tm[WT] CFUs were significantly reduced in the liver and spleen of these mice (Fig. 5e). These results indicate that immunization with apyrase-expressing bacteria confers improved protection from *S*.Tm systemic spreading.

**Protection by *S*.Tm[pApyr] depends on T-cell-dependent SIgA.** To control for effects of immunization that occur independently of adaptive immunity, we immunized recombinase-1 deficient (*Rag-1*[−/−]) mice with *S*.Tm[pApyr] or *S*.Tm[pBAD28]. In *Rag-1*[−/−] mice there was no difference in susceptibility to *Salmonella* infection between non-vaccinated or either vaccinated mouse group (Supplementary Figure 6a–e). To directly address the role of SIgA in conferring enhanced protection by vaccination with *S*. Tm[pApyr], we performed the same immunization-challenge protocol in mice with deletion of the $J_H$ gene. These *Igh*[−/−] mice cannot produce recombined variable regions of Ig heavy chains and have no detectable Ig. Analogously to *Rag-1*[−/−] mice, both non-immunized *Igh*[−/−] mice and *Igh*[−/−] mice vaccinated with either *S*.Tm transformants were equally susceptible to *Salmonella* infection (Supplementary Figure 6f-j), supporting a crucial role for SIgA in controlling the local infection and systemic spreading of the pathogen.

We carried out experiments to mechanistically link the abrogation of ATP to the loss of signaling via P2X7 on Tfh cells.

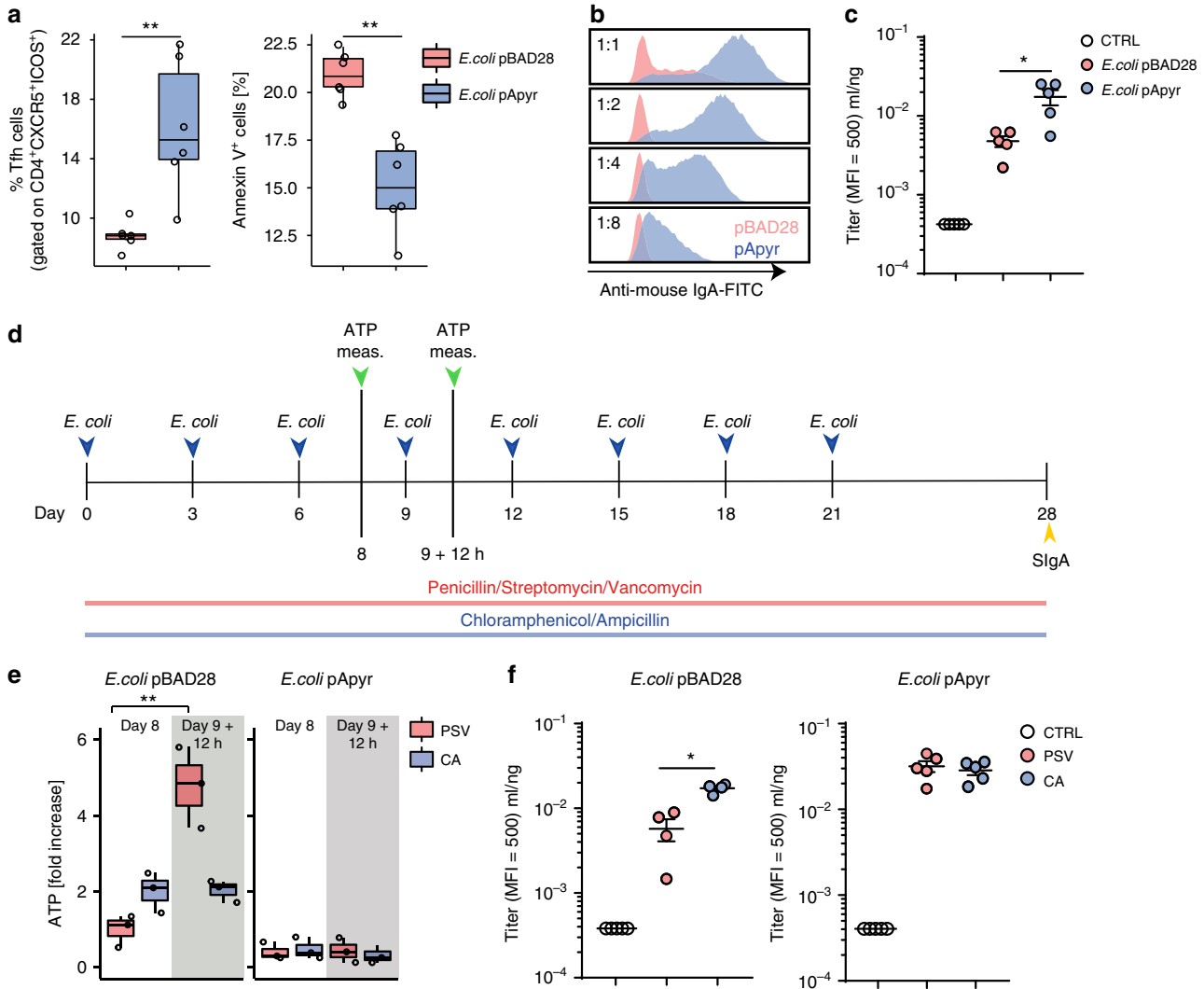

**Fig. 3** Induction of anti-*E. coli* SIgA by expression of apyrase. **a** Statistical analysis by flow cytometry of Tfh cells and Annexin V[+] cells within Tfh cells from PPs of mice immunized with *E.coli*[pBAD28] or *E.coli*[pApyr] at day 22. **b** Flow cytometry for anti-*E. coli* IgA in intestinal wash from mice immunized with *E. coli*[pBAD28] or *E. coli*[pApyr]. **c** Intestinal anti-*E.coli* IgA titer in non-immunized mice (CTRL) and mice immunized with *E. coli*[pBAD28] or *E. coli*[pApyr]. **d** Diagram showing the immunization protocol with *E.coli*[pBAD28] and *E.coli*[pApyr] in mice maintained with Chloramphenicol and Ampicillin (CA) (a bactericidal mix active on endogenous flora but not on CA-resistant *E. coli*[pBAD28] and *E. coli*[pApyr]) or Penicillin/Streptomycin/Vancomycin (PSV) (bactericidal on both endogenous flora as well as *E. coli* transformants) in drinking water. **e** Fold increase of ileal ATP in mice maintained in PSV or CA before (day 8) and 12 h after (day 9 + 12 h) orogastric gavage with the indicated transformants. **f** Intestinal anti-*E.coli* IgA titer in untreated mice (CTRL) or in response to *E. coli*[pBAD28] (left panel) or *E. coli*[pApyr] (right panel) gavaging in mice maintained in PSV or CA. The boxplots in **a** and **e** show median and upper and lower quartiles. The extreme lines show the highest and lowest value . The boxplot is overlaid with the visualization of single observations. Two-tailed Mann–Whitney *U*-tests. \*\**p* < 0.01 **a** and Kruskal–Wallis with Dunn's post-test. \**p* < 0.05, \*\**p* < 0.01 **c**, **e**, **f**. One representative experiment out of at least two is shown

We therefore immunized $P2rx7^{-/-}$ mice with $S.$Tm[pBAD28] or $S.$Tm[pApyr]. Whereas $S.$Tm[pApyr]-induced enhanced IgA responses in WT littermates, $P2rx7^{-/-}$ mice generated analogous amounts of $S.$Tm-specific SIgA after immunization with live-attenuated $S.$Tm[pBAD28] or $S.$Tm[pApyr] (Supplementary Figure 7a). Accordingly, $P2rx7^{-/-}$ mice were equally protected from local and systemic infection with virulent $S.$Tm irrespective of the immunization strain (Supplementary Figure 7b–d).

**Improved response to inactivated oral vaccines by apyrase**. We tested the ability of apyrase to enhance the induction of IgA by inactivated oral vaccines[3,24] (Supplementary Figure 8a). We generated these vaccines by treating pure-cultured bacteria with 1% peracetic acid (PA)—a strong oxidizing agent. Notably, no

increase of extracellular ATP was detected in the intestine of GF mice upon oral administration of inactivated bacteria (Supplementary Figure 8b). The oxidative treatment also abolished the function of the apyrase enzyme, resulting in identical IgA priming to a vaccine constructed from the empty-vector-carrying strain (Fig. 6a). Correspondingly, similar protection from infection with virulent $S.$Tm was observed with both vaccines (Fig. 6b, c). We therefore administered inactivated $S.$ Typhimurium with or without crudely-purified apyrase extracted from $E.$ $coli$ that significantly decreases eATP in the small intestine (Supplementary Figure 8c). Inclusion of apyrase in the vaccine preparation greatly improved the IgA titer induced, similarly to that induced by infection with the live-attenuated apyrase-expressing strain (Figs. 4d, 6d), and generated enhanced protection from challenge with virulent $S.$Tm (Fig. 6e, f). It was previously observed that the

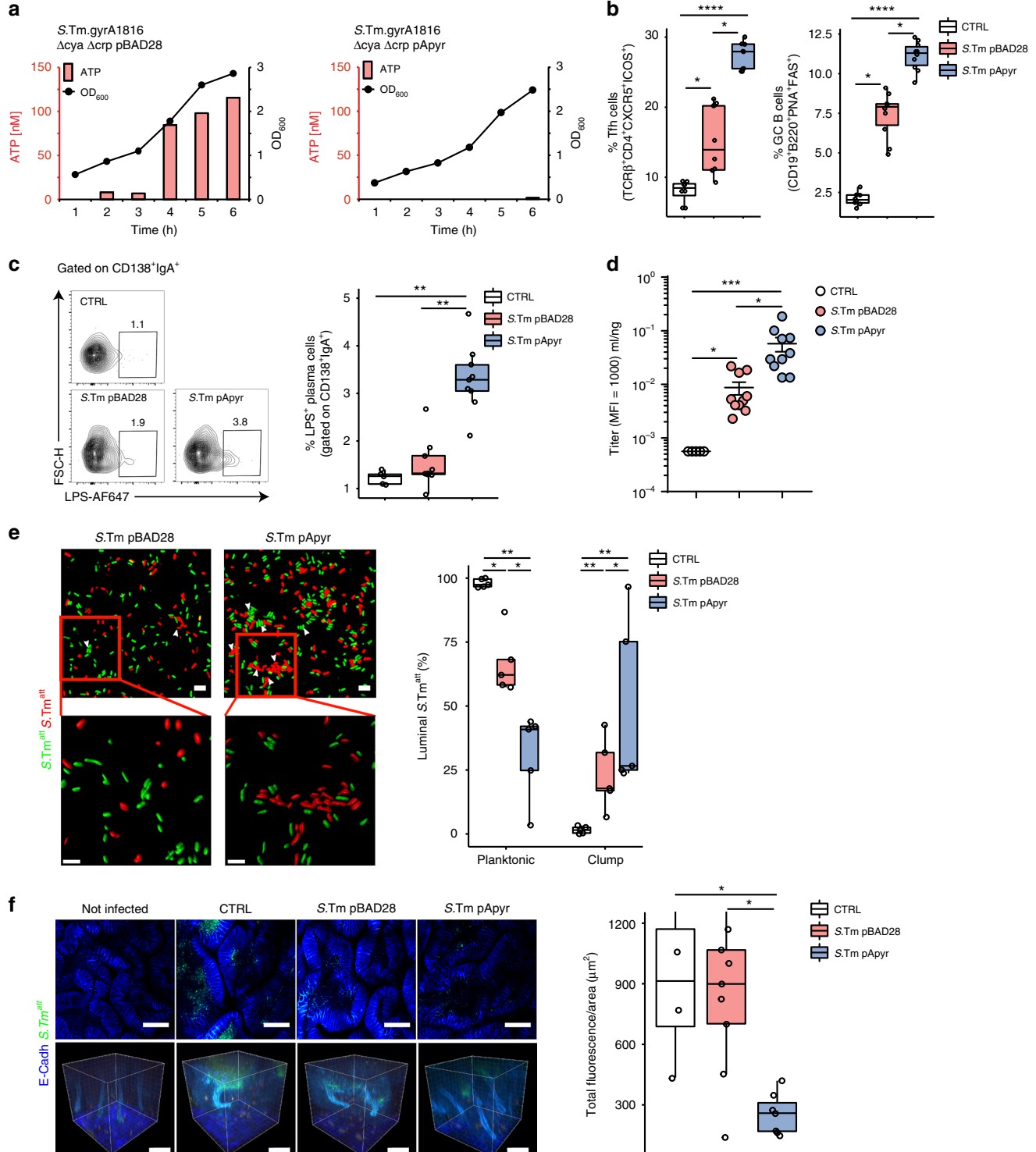

**Fig. 4** Enhanced anti-*Salmonella* SIgA by apyrase. **a** ATP concentrations in culture medium (bars) and bacterial growth (OD$_{600}$) over time for *S*.Tm$^{pBAD28}$ and *S*.Tm$^{pApyr}$. **b** Quantification of Tfh and GC B cells and **c** representative contour plots with statistical analysis of plasma cells specific for *S*.Tm LPS in PPs 48 h after orogastric infection with *S*.Tm$^{WT}$ in non-immunized mice (CTRL) and mice immunized with *S*.Tm$^{pBAD28}$ or *S*.Tm$^{pApyr}$. **d** Intestinal anti-*S*.Tm IgA titer in non-immunized mice (CTRL) and mice immunized with *S*.Tm$^{pBAD28}$ or *S*.Tm$^{pApyr}$. **e** Representative 2D (upper panels, scale bar: 5 μm) and 3D (lower panels, scale bar: 5 μm) images, and statistical analysis of bacterial clumping in live cecal content from mice vaccinated with *S*.Tm$^{pBAD28}$ or *S*.Tm$^{pApyr}$, 8 h after orogastric administration of a 1:1-mix of mCherry- and GFP-tagged *S*.Tm$^{att}$ (10$^7$ CFU). CTRL, non-vaccinated mice. **f** Images from two-photon confocal microscopy of caeca and 3D rendering of crypts from E-cadherin-mCFP mice 18 h after infection with 10$^7$ CFU of GFP-tagged *S*.Tm$^{att}$. Infected mice were either non-vaccinated (CTRL) or vaccinated with the indicated *S*.Tm transformant. Graph shows the statistics of total fluorescence inside the crypts per unit of internal area. The boxplots show median and upper and lower quartiles. The extreme lines show the highest and lowest value . The boxplot is overlaid with the visualization of single observations. Kruskal–Wallis with Dunn's post-test. *$p < 0.05$, **$p < 0.01$, ***$p < 0.001$, ****$p < 0.0001$. One representative experiment out of at least three is shown

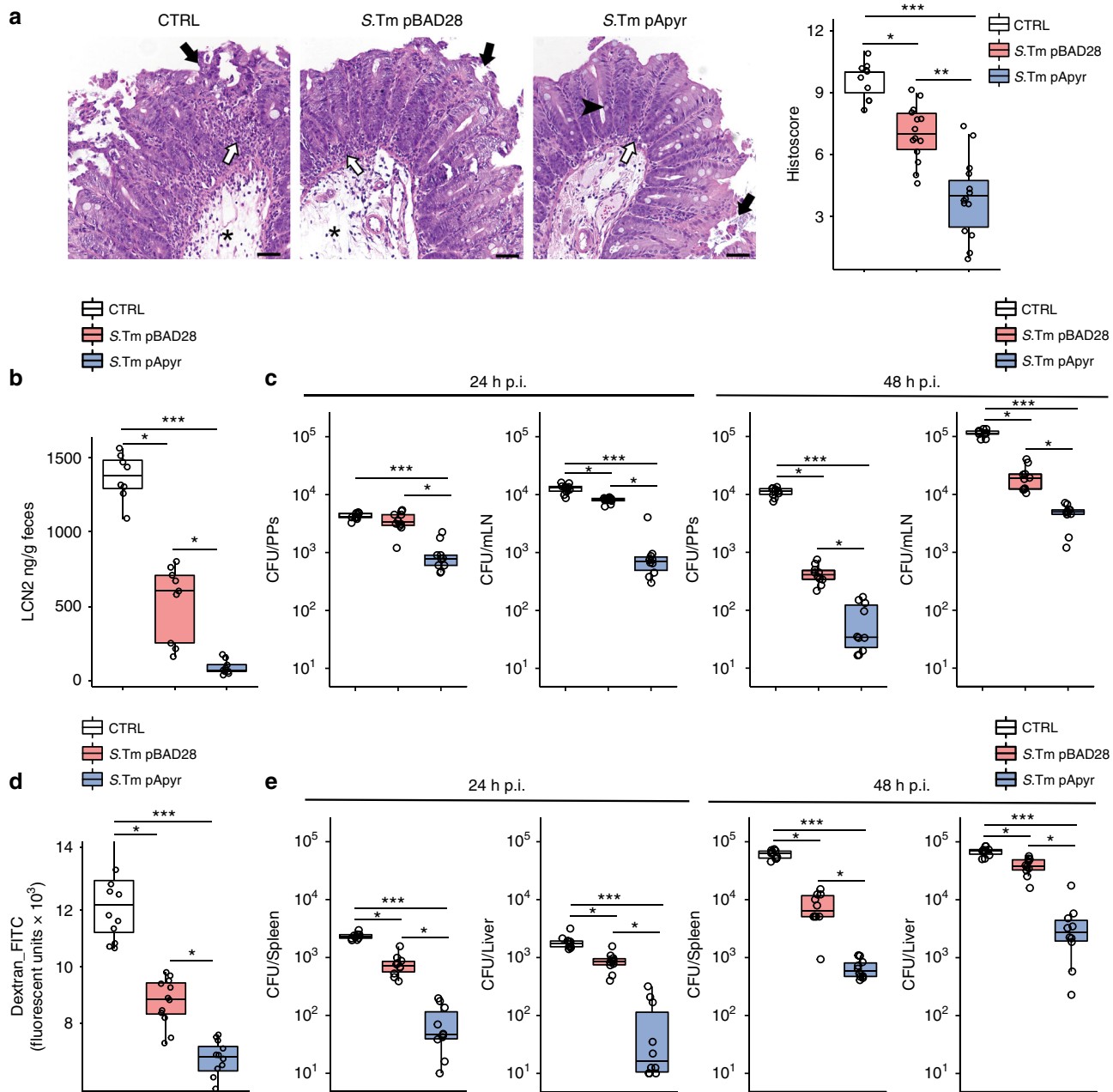

**Fig. 5** Enhanced protection from *Salmonella* infection by vaccination with *S*.Tm[pApyr]. Wild-type mice were gavaged with PBS (CTRL) or vaccinated with *S*.Tm[pBAD28] and *S*.Tm[pApyr]. Streptomycin (Sm) pretreated mice were infected with *S*.Tm[WT] and analysed 24 h and 48 h later. **a** Representative H&E sections of the cecum at 24 h post infection and statistical analysis of histopathological scores. Star: submucosal edema; white arrow: neutrophils aggregates; black arrow: epithelial defects; arrowhead: goblet cells. Scale bar: 50 μm. **b** Fecal Lipocalin 2 (LCN2) quantification 24 h post infection with *S*.Tm[WT]. **c** Pathogen loads (CFU) in PPs and mesenteric lymph nodes (mLN) 24 h (left panels) and 48 h (right panels) after infection. **d** Intestinal permeability to FITC-dextran 24 h after infection with *S*.Tm[WT] in mice gavaged with PBS or vaccinated with *S*.Tm[pBAD28] or *S*.Tm[pApyr]. Serum levels of 70-kDa FITC-dextran were assessed 4 h after gavage. **e** Pathogen loads (CFU) in spleen and liver 24 h (left panels) and 48 h (right panels) after infection. The boxplots show median and upper and lower quartiles. The extreme lines show the highest and lowest value . The boxplot is overlaid with the visualization of single observations. Kruskal–Wallis with Dunn's post-test. *$p < 0.05$, **$p < 0.01$, ***$p < 0.001$. One representative experiment out of three is shown

effectiveness of inactivated vaccines was dependent on the extent of colonization[24]. These results suggest that a dominant inhibitory effect of the microbiota is eATP production, as antibody titers induced by the inactivated vaccine in GF mice are similar to those induced by the inactivated vaccine plus apyrase in colonized animals (Fig. 6d, g). This secretory response correlates with effective protection from subsequent infection (Fig. 6e, f, h). Therefore, inclusion of recombinant apyrase into inactivated oral vaccines greatly improves the responsiveness to these preparations without any major side-effects.

## Discussion

The purinergic signaling system, which uses ATP and related nucleotides as signaling molecules, plays pleiotropic roles in regulating physiological and pathological responses in virtually all mammalian tissues[25,26]. This intercellular communication

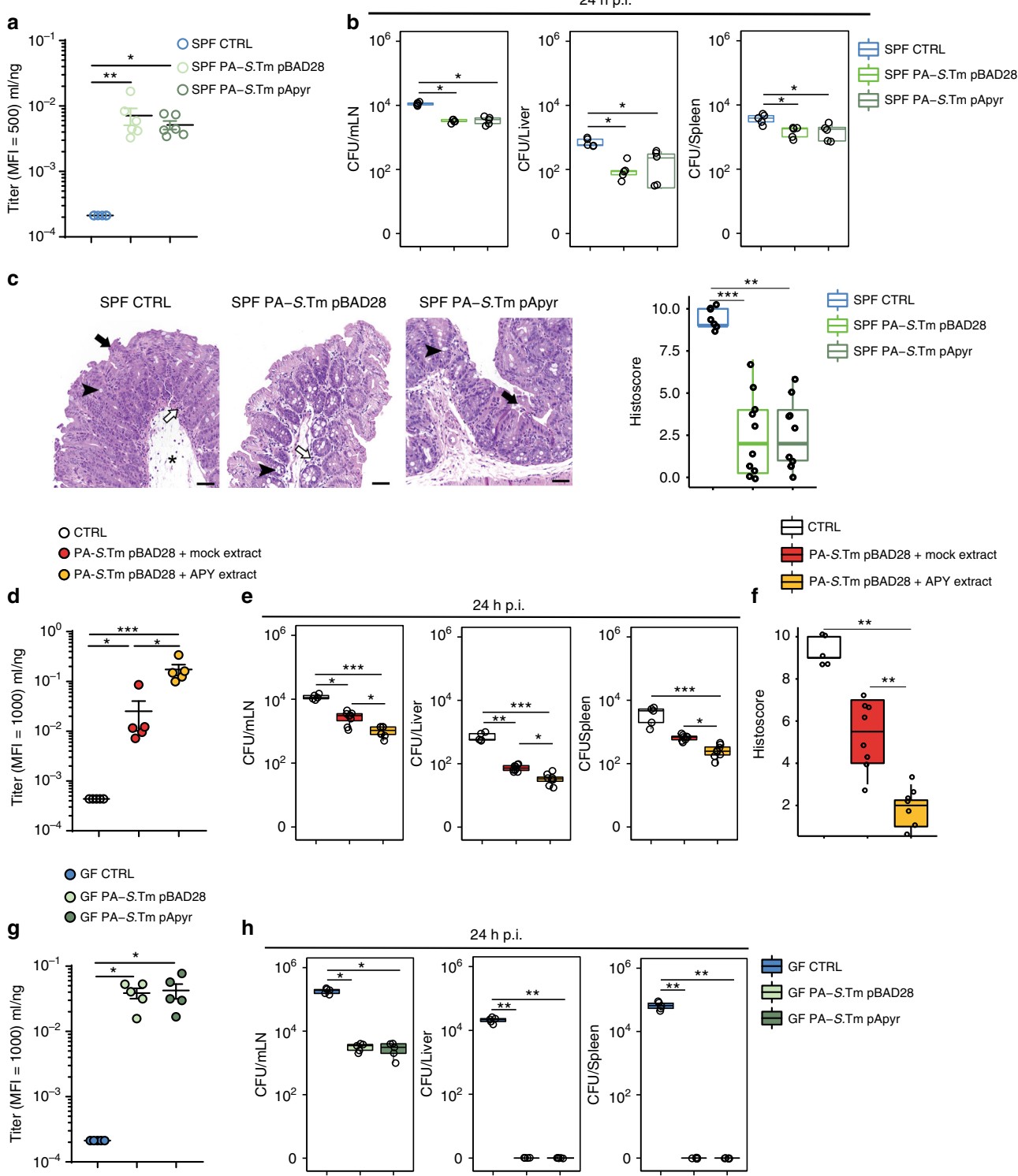

**Fig. 6** Apyrase enhances the induction of IgA by inactivated oral vaccines. SPF and GF mice were untreated (CTRL) or immunized with PA-*S*.Tm transformants together with crudely-purified apyrase (APY extract) or mock extract (mock extract) where indicated, pretreated with streptomycin, infected with *S*.Tm^WT (10^8 CFU i.g.) and analysed 24 h later. **a** Intestinal lavage IgA titer and **b** pathogen loads (CFU) in mLN, liver and spleen in SPF mice either non-immunized (CTRL) or immunized with PA-*S*.Tm^pBAD28 or PA-*S*.Tm^pApyr. **c** Representative H&E sections of the cecum from SPF mice at 24 h post infection and statistical analysis of histopathological scores. Star: submucosal edema; white arrow: neutrophils aggregates; black arrow: epithelial defects; arrowhead: goblet cells. Scale bar: 50 μm. **d** Intestinal anti-*S*.Tm IgA titer, **e** pathogen loads (CFU) in mLN, liver and spleen and **f** statistical analysis of histopathological scores in non-immunized mice (CTRL) and mice immunized with PA-*S*.Tm^pBAD28 conditioned with the indicated extract. **g** Intestinal lavage IgA titer and **h** pathogen loads (CFU) in mLN, liver and spleen in GF mice either non-immunized (CTRL) or immunized with PA-S.Tm^pBAD28 or PA-S. Tm^pApyr. The boxplots show median and upper and lower quartiles. The extreme lines show the highest and lowest value . The boxplot is overlaid with the visualization of single observations. Kruskal–Wallis with Dunn's post-test. *$p < 0.05$, **$p < 0.01$, ***$p < 0.001$. One representative experiment out of two is shown

modality emerged very early in evolution. In the endosymbiotic relationship between α-proteobacteria and the archeon, from which the eukaryotic cell originated, ATP released by mitochondria (i.e. α-proteobacteria) evolved also as a signaling molecule to communicate mitochondrial fitness[27]. Bacteria release ATP via mechanosensitive channels[28,29]. Proteobacteria, in particular, were shown to secrete elevated amounts of ATP that were modulated by adherence to different surfaces[30]. Indeed, ATP release as well as ATP breakdown enzymatic systems are present in all kingdoms of life[26] (Supplementary Figure 2a). The observation that ATP released by bacteria limits Tfh cells abundance via P2X7 receptor and ensures controlled T-dependent SIgA responses in the small intestine indicates eATP can act as an inter-kingdom signalling molecule. This regulatory pathway plays a crucial role in shaping a beneficial gut ecosystem for host metabolism[6]. On the other hand, it contributes to the relative resistance of the intestinal adaptive immune system to generating high-affinity SIgA upon oral immunization. Overcoming this insensitivity has typically required the use of live-attenuated oral vaccines, which encompass significant safety risks. Transient subversion of bacterial ATP-mediated control of Tfh cells by apyrase could be exploited to enhance T cell-dependent IgA induction by inactivated oral vaccines or to limit the harmful potential of live-attenuated oral vaccines by specifically enhancing SIgA response and limiting intestinal inflammation, at the same time eliciting more effective and protective SIgA against enteropathogens. Our experiments demonstrate that this strategy is safe and generates potent protective immune responses.

## Methods

**Mice**. Animal experiments were performed in accordance with the Swiss Federal Veterinary Office guidelines and authorized by the Cantonal Veterinary. C57BL/6J, $P2rx7^{-/-}$ (B6.129P2-$P2rx7tm1Gab$/J), $Rag1^{-/-}$, $Jh^{-/-}$, and E-cadherin-mCFP (B6.129P2(Cg)-Cdh1tm1Cle/J) mice were bred in the SPF facility at the Institute for Research in Biomedicine, Bellinzona, Switzerland. C57BL/6J GF mice were maintained in flexible film isolators at the Clean Animal Facility, University of Bern, Switzerland. Where indicated, mice were treated with an antibiotic association containing Metronidazole (2.5 mg), Ampicillin (2.5 mg), and Vancomycin (1.25 mg) (VAM) in 200 µl per mouse by oral gavage. For analysis of epithelial turnover, mice were starved for 36 h with bedding chips and drinking water. After starvation, mice were re-fed for 24 h and then sacrificed for analysis. The re-feeding period was set to begin at 8:00 in all experiments. Where indicated, mice were pretreated for 15 day with VAM. In all experiments, up to five mice were housed per cage in a 12-h light-12-h dark cycle.

**Determination of the ileal volume**. To calculate the ileal volume, the last 5 cm of the terminal ileum were excised from 8-week-old-female mice and fixed in neutral buffered formalin (16 h at 4 °C). After fixation samples were dehydrated (70% ethanol, two changes, 1 h each; 80% ethanol, one change, 1 h; 95% ethanol, one change, 1 h; 100% ethanol, three changes, 1.5 h each; xylene, three changes, 1.5 h each), embedded in paraffin and then cut at microtome to the desired thickness (6 µm). Ten sections spanning the 5 cm were obtained and stained with hematoxylin and eosin (H&E). The internal areas of sections were calculated by ImageJ and the mean value (base) multiplied for 5 cm (height)[31] (Supplementary Figure 9).

**Quantification of ATP**. For quantification of ileal ATP, intestinal content was collected by lavage with 10 ml of intestinal wash buffer (PBS, 0.5 M EDTA, Soybean trypsin inhibitor, PMSF), spun and filtered (0.22 µm) to remove any bacteria-sized contaminants and immediately frozen in dry ice. ATP concentration in the intestinal washes was multiplied for the dilution factor to obtain the actual endoluminal ATP concentration (Supplementary Figure 9). Bile and urine were collected from gallbladder and bladder through puncture with a 34G needle. For quantification of ATP secreted by commensal bacteria in culture, intestinal content was plated on BHI agar and cultured for 16 h at 37 °C. Single colonies were picked and cultured in BHI broth or LB. To quantify the ATP production during bacterial growth, the bacterial culture supernatant was collected at different O.D., centrifuged and filtered (0.22 µm). For quantification of ATP in serum, inferior caval, jugular and portal veins, and hearth were exposed and blood collected through puncture with a 34G needle. Emolysed sera were discharged. The extracellular ATP concentration was evaluated by bioluminescence assay with recombinant firefly luciferase and its substrate D-luciferin according to the manufacturer's protocol (Life Technologies Europe B.V.).

**OMVs isolation**. OMVs were isolated from 250 ml of LB cultures[32]. E. coli[pBAD28] and E. coli[pApyr] were grown to late-exponential phase (OD$_{600}$ ~0.8–1.0) and removed from culture supernatants by centrifugation. The collected supernatants were filtered (0.22 µm) and concentrated using the Vivaspin 20 concentrators, molecular weight cutoff 50-kDa (GE Healthcare), to eliminate free apyrase (~27 kDa) from the medium. The collected concentrates were then centrifuged at 270,000×g for 3 h at 4 °C to yield crude OMVs preparations that were resuspended in PBS and tested for apyrase activity.

**Apyrase activity test**. To test the apyrase activity in bacterial supernatants, OMVs and intestinal washes, samples were incubated with 50 µM ATP for 30 min at room temperature. ATP concentration was evaluated by a bioluminescence assay with recombinant firefly luciferase and its substrate D-luciferin according to the manufacturer's protocol (Life Technologies Europe B.V.). The ATPase activity of the samples was expressed as the percentage of non-degraded ATP.

**Antibodies and flow cytometry**. The following mAbs were purchased from BD Biosciences: biotin conjugated anti-CXCR5 (clone: 2G8, Cat.#: 551960, dilution 1:50), phycoerythrin (PE) conjugated anti-ICOS (clone: 7E.17G9, Cat.#: 552146, dilution 1:200), PE conjugated anti-CD138 (clone: 281-2, Cat #:553714, dilution 1:100) and PE conjugated anti-Fas (clone: Jo2, Cat.#: 554258, dilution 1:200). Allophycocyanin (APC) conjugated anti-B220 (clone: RA3-6B2, Cat.#: 103212, dilution 1:200), APC-Cy7 conjugated anti-CD19 (clone: 6D5, Cat.#: 115530, dilution 1:200), PE-Cy7 conjugated anti-CD4 (Clone: GK1.5, Cat.# 100422, dilution 1:200), Alexa Fluor488 anti CD326 (Ep-CAM) (clone: G8.8, Cat.#: 118210, dilution 1:200) and APC conjugated streptavidin (Cat.#: 405207, dilution 1:200) were from Biolegend. Peridinin chlorophyll protein (Percp)-eFluor710 conjugated anti-CD3 (Clone: 17A2, Cat.#: 46-0032-80, dilution 1:200), PE conjugated anti-Ki-67 (clone: SolA15, Cat.#: 12-5698-82, dilution 1:200) and APC-eFluor780 anti-CD45.2 (clone: 104, Cat.#: 47-0454-82, dilution 1:200) were obtained from eBioscience. Fluorescein labelled Peanut Agglutinin (PNA) (Cat.#: FL-10-71, dilution 1:500) was purchased from Vectorlabs. Fluorescein Isothiocyanate (FITC) conjugated anti-IgA (Cat.#: 1040-02, dilution 1:500) was obtained from Southern Biotech. To quantify IgA plasma cells specific for S.Tm, we labeled S.Tm LPS (Sigma-Aldrich) with Hydrazide-Biotin reagent (Pierce Biotechnology) according to the manufacturer's instructions. Cells were stained with 30 µg ml$^{-1}$ biotinylated LPS, FITC-anti-IgA, PE-anti-CD138 antibodies at 4 °C for 45 min and then with Alexa Fluor 647 labeled streptavidin[33]. Annexin V staining was performed in Biolegend Annexin V binding buffer (Cat.#: 422201) containing Annexin V APC (dilution 1:300) at a cell density of $1 \times 10^6$ cells ml$^{-1}$. Samples were washed twice in the same buffer, acquired on LSRFortessa flow cytometer (BD Biosciences) and data analysed using FlowJo software (TreeStar, Ashland, OR) or FACS Diva software (BD Biosciences).

**Bacterial strains and growth conditions**. Full length $phoN2$::HA fusion, encoding periplasmic ATP-diphosphohydrolase (apyrase) of $Shigella flexneri$, was cloned into the polylinker site of plasmid pBAD28 (ATCC 8739387402), under the control of the P$_{BAD}$ L-arabinose inducible promoter, generating plasmid pHND10[12]. Bacteria transformed with pBAD28 or pHND10 were grown in LB medium supplemented with L-arabinose (0.03%) and ampicillin (100 µg ml$^{-1}$). For infection experiments, S.Tm$^{WT}$ (SL1344 wild-type clone SB300) or the respective mutants were cultured in LB containing the appropriate antibiotics for 12 h at 37 °C, diluted 1:20 and sub-cultured for 3 h in 0.3 M NaCl supplemented LB without antibiotics. The bacterial strains used in the study are listed in Supplementary Table 1.

**Determination of specific antibody titers by flow cytometry**. Specific antibody titers in mouse intestinal washes were measured by flow cytometry[34]. Intestinal contents were collected by lavages with 5 ml of intestinal wash buffer (PBS, 0.5 M EDTA, Soybean trypsin inhibitor, PMSF), spun and filtered (0.22 µm) to remove any bacteria-sized contaminants. Bacterial targets were resuspended at a density of $10^7$ bacteria ml$^{-1}$. Intestinal washes were serially diluted and 25 µl of each dilution were incubated with 25 µl of bacterial targets suspension at 4 °C for 1 h. After two washes, bacteria were incubated for 1 h with monoclonal FITC-anti-mouse IgA and then resuspended in 2% paraformaldehyde in PBS for acquisition on a FACSCanto using FSC and SSC parameters in logarithmic mode. ELISA was used to determine the total IgA concentration in an undiluted aliquot of the same intestinal wash sample used for analysis in flow cytometry. Median fluorescence intensities (MFI) were plotted against antibody concentrations for each sample and 4-parameter logistic curves fitted using Prism (Graphpad, La Jolla, CA). Titers were calculated from these curves as the inverse of the antibody concentration giving an above-background signal. The concentration of total antibody titer required to achieve a given MFI (for example = 500) was calculated by re-arrangement of the fitted 4-parameter logistic equation for each samples. As this value is low where a strong antibody response is present, the inverse of this value was plotted. Thus titers are calculated as the inverse total antibody concentration required to achieve a given MFI. The y-axis value chosen as "above background" necessarily varies between experiments due to the flow cytometer settings, but is constant within any one analysis[34].

**Treatment of bacterial cultures with antibiotics**. Ampicillin ($2.5\,\mu g\,ml^{-1}$), vancomycin ($1\,\mu g\,ml^{-1}$), metronidazole ($1\,\mu g\,ml^{-1}$) were added to intestinal bacterial culture when $OD_{600}$ reached 0.5 value. At different times after addition of antibiotics, bacterial cultures were spun and supernatants collected in a sterile tube. ATP concentration was evaluated by bioluminescence assay in filtered supernatants (see above).

**Production of peracetic acid killed vaccines**. To produce PA killed vaccines, bacteria grown for 16 h to late stationary phase were collected by centrifugation and resuspended at a density of $10^9$–$10^{10}\,ml^{-1}$ in sterile PBS. Peracetic acid (Sigma-Aldrich) was added to a final concentration of 1% and after vigorous mixing, the suspension was incubated for 60 min at room temperature. After extensive washing, the final pellet was resuspended at a concentration of $10^{11}$ particles $ml^{-1}$ in sterile PBS[24]. Each batch of vaccine was tested before use in order to confirm absence of live bacteria.

**Preparation of periplasmic extract**. *E. coli*[pBAD28] and *E. coli*[pApyr] were grown as described above and collected by centrifugation. After washing, bacteria were resuspended ($10^{10}$ CFU $ml^{-1}$) in PBS with 30 mM Tris-HCl (pH 8.0), 4 mM EDTA, 1 mM PMSF, 20% sucrose and 0.5 mg $ml^{-1}$ lysozyme and incubated 2 min at 30 °C. $MgCl_2$ (10 mM final) was added to the bacterial solution and incubation was continued for 1 h at 30 °C. At the end of the incubation period bacterial suspensions were centrifuged at $11,000\times g$ for 10 min at 4 °C and supernatants were stored (periplasmic extract).

**Oral vaccination protocols**. For vaccination with *E. coli* transformants, *E. coli*[pBAD28] and *E. coli*[pApyr] were collected by centrifugation, washed in sterile PBS and $10^{10}$ CFUs administered to mice by orogastric gavage. The procedure was repeated every 3 days for 3 weeks and mice were sacrificed at day 22 or 28 (Supplementary Figure 3b). For vaccination with S.Tm transformants, S.Tm[pBAD28] and S.Tm[pApyr] were collected by centrifugation, washed in sterile PBS and $5\times10^9$ CFUs administered to mice by orogastric gavage. The procedure was repeated every 3 days for three times (Supplementary Figure 5b). The pBAD promoter is constantly active in the gut lumen (E.S. and W.D.H., unpublished observations), however, to ensure optimal apyrase expression by administered S.Tm transformants, mice were maintained with 0.3% arabinose in drinking water. On day 28, mice were used for infection experiments. For oral vaccination with PA killed S.Tm[pBAD28] and S. Tm[pApyr], mice received $10^{10}$ particles of the respective transformants in PBS every 3 days for three times. When periplasmic extracts from *E. coli*[pBAD28] (mock extract) or *E. coli*[pApyr] (APY extract) were used, 100 μl of the extract was orally gavaged every 12 h starting three days before the first day of immunization until day 10. On day 28, mice were used for infection experiments (Supplementary Figure 8a).

**Challenge infections with *S. Typhimurium***. Mice were pretreated with $1\,g\,kg^{-1}$ streptomycin sulfate in sterile PBS by gavage. Twenty-four hours later, S.Tm[wt] ($10^8$ CFUs 0.1 $ml^{-1}$ PBS) were gavaged into the stomach. For determination of total bacterial loads, homogenates of PPs, mLN, spleen and liver collected at 24 and 48 h after infection, were plated on MacConkey agar plates containing $50\,\mu g\,ml^{-1}$ streptomycin.

**Live confocal microscopy of cecal content**. Vaccinated or control mice were pretreated with $0.8\,g\,kg^{-1}$ ampicillin sodium salt in sterile PBS by gavage. Twenty-four hours later, mice received $10^7$ CFUs of a 1:1 mix of mCherry-(pFPV25.1) and GFP-(pM965) expressing avirulent S.Tm. For imaging, cecum content was gently diluted to 1:10 w/v in sterile PBS containing 6 μg $ml^{-1}$ chloramphenicol, avoiding heavy mixing[3]. The suspension (200 μl) was transferred to 35 mm dish, 14 mm glass diameter, poly-D-lysine coated Petri dish (MatTek Corporation) and imaged using a Leica TCS SP5 confocal microscope with a ×100/1.44 NA oil immersion objective (HCX PL APO CD ×100/1.44 oil). Individual bacteria were visually scored as planktonic, whereas aggregates of equal or more than three bacteria were scored as clumps.

**Multiphoton microscopy and analysis**. Vaccinated or control E-cadherin-mCFP mice were pretreated with $0.8\,g\,kg^{-1}$ ampicillin sodium salt in sterile PBS by gavage. After 24 h, mice received $10^7$ CFUs of S.Tm[att] expressing GFP constitutively. After 18 h, the infected animals were sacrificed and the whole-ceca collected for 2-photon analysis. Deep tissue imaging was performed on a customized two-photon platform (TrimScope, LaVision BioTec)[35]. The objective used was a Nikon Apo LWD ×25/1.10W IR Corrected. The fluorescence signal has been separated using a custom configuration of detecting PMTs equipped with a set of dichroic mirror and selective bandpass filters for the Blue, Green, and Red channels (respectively detecting the fluorescence in the range of 450 nm–495 nm, 500 nm–550 nm, 600 nm–635 nm). 3D reconstructions of the whole crypts shown in Fig. 4f and Supplementary Figure 10 were performed by acquiring a z-stack with z-step between slices of 3 μm, for a total depth of 120 μm. Images were analysed using FIJI software[36] with a custom-developed macro to automate image processing: the internal area of crypts was segmented by applying a threshold on the intensity of

the red autofluorescence (intensity above $I = 1500$ a.u.). Within the identified ROI (in the red channel) the green particles above the threshold (intensity above $I = 5000$ a.u. in the green channel) were detected and total fluorescence was measured by summing the single particles fluorescence intensity along the entire z-stack. The obtained total intensity was normalized by the total internal area of the crypts (obtained by summing up the single slice areas found in the red channel). 3D renderings shown in Fig. 4f and Supplementary Figure 10 were made with Clear Volume plugin of FIJI software[37].

**Histological evaluation of *Salmonella*-induced typhlitis**. Ceca from all animals were examined at necropsy, fixed in 10% neutral buffered formalin for at least 48 h prior to embedding in paraffin and stained with H&E. Pathological scores were determined in a blinded manner using a scoring scheme, which takes into consideration the severity of submucosal edema (scores 0–3), neutrophilic infiltration into the lamina propria (scores 0–4), loss of goblet cells (scores 0–3), and epithelial damage (scores 0–3)[17,38].

**Phylogenetic tree of apyrases**. Protein sequences were retrieved from GenBank by searching for the terms "apyrase" and "ectonucleoside triphosphate diphosphohydrolase". Putative and partial sequences were excluded from the analysis. Protein sequences were aligned using MUSCLE[39]. The phylogenetic trees were inferred by using FastTree2[40].

**Statistical analysis**. Where two groups of data were compared, analysis was carried out using two-tailed Mann–Whitney U-tests. Where more than two groups were compared, data were analyzed by Kruskal–Wallis test with Dunns post-test to account for multiple testing. Statistical tests were performed using GraphPad Prism 7.02 for Windows (http://www.graphpad.com).

**Reporting Summary**. Further information on experimental design is available in the Nature Research Reporting Summary linked to this Article.

## Data availability
The authors declare that the main data supporting the findings of this study are available within the Article and its Supplementary Information files. A Reporting Summary for this Article is available as a Supplementary Information file. Extra data are available at 10.5072/zenodo.254203.

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

## Acknowledgements

We thank Michela Perotti, Tanja Rezzonico-Jost, and Sara Maffei (Institute for Research in Biomedicine, Bellinzona) for help with mice experiments. The work was supported by grant 310030_159491 (to F.G.) and 310030B_173338 (to W.D.H.) of the Swiss National Science Foundation (SNSF).

## Author contributions

F.G., L.P. and M.P. designed experiments; L.P. and M.P. performed most experiments with contribution by F.S. and M.R.; D.S. and M.N. generated *E. coli* and *Salmonella* transformants; G.P. performed histopathological analyses; R.D'A. and S.F.G. performed two-photon microscopy; M.T. performed confocal microscopy; W.D.H. and E.S. provided fluorescent bacteria and experimental design for agglutination experiments and immunization with peracetic acid-inactivated *Salmonella*. F.G. supervised the work. F.G. and E.S. wrote the manuscript.

## Additional information

**Competing interests:** M.P. and F.G. are listed as inventors in a patent application that covers the method described in this Article (WO2017108935A1). The remaining authors declare no competing interests.

