## [Peer Review File · Nature Communications]

Reviewers' comments:

Reviewer #1 (Remarks to the Author):

The work of Proietti and colleagues describes an interesting novel observation with potentially broad implications for the development of vaccines against enteric infections. The group found that ATP released by intestinal microbes affects a subpopulation of T follicular helper cells (T_{fh}) important for the production of IgA. The work extends the earlier observation of the group that intestinal ATP levels affects viability of T_{fh} and by this the secreted IgA. The present manuscript reports that the intestinal microbiome is responsible for ATP release and by this modulates the IgA levels in the intestinal lumen. A vaccination strategy was tested that made use of apyrase expressing attenuated Salmonella, resulting in decreased ATP release. Vaccination with this strain led to increased Salmonella specific IgA titers and protection against challenge infections. There are several technical and conceptual issues that will need further attention.

Specific comments:

Formally, the statement that the released ATP is bacterial origin is not sufficiently supported. Since archaeal and eukaryotic microbes are as well members of the intestinal microbiome of SPF mice, the ATP detected in ileal lumen (Fig. 1) could be as well of archaeal, fungal or protozoan origin. The authors only follow release up release by bacteria but not by other microbes in Fig. 1C. How can the effect of purified recombinant apyrase in contrast to apyrase expressed by attenuated bacteria be explained? Apyrase expressed by recombinant bacteria will act in the periplasm and thus will result in cell-autonomous reduction of ATP release by individual bacteria. In contrast purified apyrase will act on the overall level eATP, without an effect on individual bacteria. Did the authors check if the overexpression of apyrase in *E. coli* and *S. Typhimurium* leads to release of apyrase into the culture supernatant or intestinal lumen?

Fig. 2: the models for ATP release in panel A and B are over-simplified. There is no way ATP will directly cross inner and outer membrane. For outer membrane passage, one might consider porins. However, inner membrane to periplasm passage of ATP is very unlikely, apart from situation of membrane damage or lysis of cells.

The authors show that application of purified apyrase or apyrase producing bacteria results in increased IgA titers. Can this effect be counteracted by direct application of ATP?

Fig 3E: Either the scale bars are incorrect, or STM pBAD28 is twice the size as STM pApyr. It seems that the detail sections are incorrectly selected. What defines planktonic and clump? My interpretation of this experiment is that lower amounts of bacteria are present in the lumen of STM pBAD28 vaccinated mice.

Fig. 3F: What defines the difference between STM pBAD28 and STM pApyr? The distribution appears rather similar. The right panel remains cryptic without further explanation of the approach of analysis. Use additional Suppl. Fig. to explain what was quantified here.

The way the CFU counts are presented in Fig 4 B, E, H and Suppl. Fig 2 and 3 makes it difficult to estimate the effect of the vaccination regimes. It is more appropriate to display the CFU counts with log scale Y-axes. The data shown do not justify statements such as '...generated enhanced protection' since difference between mock and apyrase or vector and pApyr are rather minute.

Minor comments:

The authors used flow cytometry, not FACS in their experiments.

Suppl. Fig. 1e: is total CFU/g feces, total amount of *E. coli*, CFU of plasmid harbouring *E. coli* given here?

For Fig 2A and B, please indicate the time point of induction of Apy in the growth experiments.

Reviewer #2 (Remarks to the Author):

Thank you for the opportunity to review this interesting and generally well-written paper.

The authors have consistently shown a link between the ectopic (i.e. introduced non-pathogen or pathogen) expression of ATP-diphosphohydrolase which reduces extracellular ATP levels, and a significant biological phenotype.

These studies were conducted to prove a novel and attractive hypothesis that it is microbiome-released ATP that drives reductions in IgA levels, to preserve the microbiome. The extension of this hypothesis is that vaccination impact can be enhanced if the microbiome produced eATP is reduced.

The evidence that there is biologically-relevant levels of microbiome-produced eATP is relatively weak and there are other explanations for the eATP observed. On balance they show that depletion of eATP increases IgA but the attempts to show it is this increased IgA that is responsible for a biological/vaccination effect on infection (e.g. 3E Clump data, the responses of 5 mice were bimodal) are less convincing. The phenomenon described may well be mediated through P2rx7, if they occur.

Specific issues

On a general note, the choice of axis units on the graphs seems to vary within experiments and between timepoints, making comparisons more complicated than necessary (e.g. Supp Figure 2E). I would also suggest that the Methods and Legends carry too little detail - in some cases to interpret the data, in others to allow reproduction of the data.

The source of ATP in the natural putative phenomenon of eATP regulation of IgA levels needs further exploration. There are gross differences between the GI tracts from normal animals, and those from animals lacking a microbiome. While it is possible that the ATP might come directly from bacteria, it is also possible the host homeostatic mechanisms used to control the inflammation that might be driven by the microbiome, also generate extracellular ATP. Is there a means of removing the microbiome over an extended period, using the antibiotic cocktail described, i.e. from an adult mouse, and correlating ATP reductions with loss of this microbiome, and an increase in IgA?

Could the changes (Figure 1F) generated at 3h post VAM treatment have been generated by ATP released from host cells responding to the increase in PAMPs released from the bacteria killed by the antibiotics, i.e. instead of, or possibly as well as, the ATP derived from bacterial lysis?

The authors suggest that a reduction in eATP very quickly converts into enhanced specific IgA production. How quickly is this phenomenon occurring and would there be sufficient time to fully engage the bacterium-specific B cells, Thf etc. to drive an IgA response, after eATP is reduced through ectopic expression? It is not clear when (after immunization) these responses (e.g. Figures 2C and 2D) were measured. In situ, the relative timing of eATP regulation and microbiome-specific IgA 'suppression' by eATP is a key issue. Hypothetically, does it occur early, or with adapted microbiomes only?

The Salmonella immunization experiments are somewhat hard to interpret. The ability to understand the nature of the experiments is somewhat hampered by the lack of description of the strains - by searching, it is clear that the ATCC 53648 is a Salmonella enterica var Typhimurium

known as Chi4064, an attenuated *cya/crp* deletion mutant of *S. Typhimurium* SR-11, developed by Roy Curtiss. At 28 days after 3 vaccinations, 3 days apart, the Chi4064-vaccinated mice were challenged with standard *S. Typhimurium* SL1344, a more common strain of *S. Typhimurium*. Many previous studies would suggest that mice vaccinated for 1 month with attenuated *Salmonella* mutants like Chi4064 would have had vaccination-derived bacteria still in the tissues, making it more difficult to identify vaccine- and challenge-derived bacteria when estimating challenge effects. The authors conclude that the immunity observed is due to IgA, but it could be to other immune responses such as provided by CD4+ T cells; the Rag1 mutation might equally impact these other cell types. It cannot safely be assumed that the apparent reductions in bacterial counts in the immunised mice are driven by differentially heightened IgA responses, absent some more targeted analysis. The use of mCherry and GFP-tagged bacteria and photon microscopy is an interesting means of dissecting the immune response, but the genotype of the 'att' strain is cryptic, and the description of this experiment is too limited to understand the experimental design and interpretation.

From a technical perspective there are some minor issues.

1. The use of an arabinose-dependent promoter for ectopic expression of the *phoN2* means that transcription of *phoN2* quickly ceases when arabinose is withdrawn and bacterial replication occurs - was arabinose fed to the infected mice?
2. Is the luminal eATP released from the microbiome freely basement- and cell-membrane permeable - can it reach the tissues where IgA is to be produced - lumen-administered labelled ATP might reveal this.
3. Is the enhanced level of IgA a result of increased IgA transport via the pIgR, rather than synthesis?
4. The bacterial species selected for 'specificity checking' after *E. coli* immunisation include genera where significant *E. coli* cross reactions might be expected because of genome similarity, such as *Salmonella* and *Klebsiella*, especially if a K-12 lab variant was used (these are usually LPS-deficient) - are the elevated anti-bacterial responses displayed in 2C and 2D LPS specific? This is easily tested. Lastly, the species selected for comparison (2E) do not necessarily form part of the murine microbiome; were the specific isolates tested derived from their animal house and was a microbiome analysis conducted to ensure that the strains tested were relevant?

Reviewer #3 (Remarks to the Author):

This manuscript builds on previous work from the same group, where they showed that extracellular eATP stimulation via the P2RX7 receptor lead to apoptotic death of Tfh cells in the Peyer's patches. In the present study, they investigate whether eATP from intestinal microbiota might therefore inhibit specific IgA responses towards intestinal bacteria.

They report that ileal bacteria produce eATP, some of which reaches the local circulation, and that treatment with antibiotics increases eATP release by intestinal bacteria, resulting in increased Tfh death and reduced IgA responses. Conversely, expression of apyrase (an ATP-degrading enzyme) in *E. coli* led to improved induction of specific IgA antibodies.

To demonstrate the potential utility of their findings, they turn to a *Salmonella* (*S.Tm*) model, and report that vaccination with an attenuated *S.Tm* strain expressing apyrase again resulted in increased production of high affinity secretory IgA that confers enhanced protection from challenge with virulent *S.Tm*. They also show that administration of apyrase extract during vaccination with inactivated *S.Tm* leads to enhanced protective IgA responses. This study reveals a novel pathway

in which protective secretory IgA responses are limited by eATP produced by intestinal bacteria and shows that blockade of this pathway by apyrase can improve the response to oral vaccination.

Overall, this is a very interesting and high-quality study that is clearly presented, well controlled and with conclusions that are strongly supported by the data.

I have a few specific comments that the authors could consider to further clarify and/or strengthen their study.

1. The data in Fig.1D-G show that treatment with VAM leads to increased eATP release from bacteria both in vitro and in vivo. The in vivo increases are less pronounced and the ileal and portal vein ATP levels in the VAM-treated group (Fig.1F) look very comparable to those found in the untreated SPF mice in Fig.1A,B. These observations suggest that there was some intra-experimental variability in the ATP levels. This brings up one noticeable weakness in the manuscript, namely that the reproducibility of the findings and the number of independent experimental repeats were not stated. The figures appear to show representative experiments and often have quite low n values. Therefore, all the figure legends should include full information on how many times each finding was confirmed and whether the results shown are representative, or instead represent pooled values from multiple independent experiments.

2. The acute exacerbation of intestinal ATP levels following VAM treatment was accompanied by increased Annexin V expression by Tfh cells (Fig.1F,G). The authors suggest that, "... antibiotic treatment may negatively effect the induction of T-cell dependent intestinal immunity in these infections". However, is it not likely that VAM treatment leads to an acute increase in intestinal ATP, followed by a subsequent decrease in intestinal ATP (due to depletion of microbiota)? Therefore, with prolonged VAM treatment (more representative of a clinical scenario), one might expect to see a rebound in Tfh cells as the ATP levels decrease. Therefore, analogous to the excellent responses of GF mice to vaccination (Fig.4G,H), one might expect mice treated with VAM for a longer period to make enhanced responses to oral vaccination? Finally, in Fig.1G it would be good to include some representative FACS plots of the Tfh staining, as the magnitude of the increase seems quite modest.

3. The data in Fig.2 show that E.coli pApyr induces higher IgA responses than E.coli pBAD28. Does this correlate with decreased ileal and portal vein ATP concentrations? What about Tfh levels and Annexin V staining in the PP? It would be good to include these data - they could replace Fig.2E which could easily be moved to the Supplemental data. Did you ever try co-infection with both strains of E.coli? Presumably the presence of the E.coli pApyr strain would overcome any inhibitory effects of the E.coli pBAD28 strain?

4. Even having carefully read the methods, it is difficult to precisely understand why the IgA antibody titers are expressed as "ml/ng". I assume that the "ng" refers to the total IgA concentration, but what does the "ml" refer to?

5. I would recommend changing the scale on the histological data shown in Supplemental Fig.3A, as it makes it look like there is a trend to reduced pathology in the S.Tm pApyr vaccinated group. Having the same scale as used in the histology graph presented in Supplemental Fig.2C would emphasize the similarities across the groups.

6. In Supplemental Fig.3.F-I the presented data indicate that P2RX7^{-/-} mice are equally protected by S.Tm pBAD28 and S.Tm pApyr vaccination. However, in Fig.S3F, it looks like they make more IgA after pApyr immunization. Does this suggest some P2RX7-independent effects of eATP on sIgA responses? If so, this should be discussed by the authors.

7. The experiments in Fig.4D-F show that addition of apyrase extract results in enhanced

protection following vaccination with fixed S.Tm. This suggests that apyrase could be utilized as an adjuvant for oral vaccination. This raises the issue as to whether apyrase is immunogenic itself and are specific IgA antibodies produced against apyrase?

Reviewer #1 (Remarks to the Author):

The work of Proietti and colleagues describes an interesting novel observation with potentially broad implications for the development of vaccines against enteric infections. The group found that ATP released by intestinal microbes affects a subpopulation of T follicular helper cells (T_H) important for the production of IgA. The works extends the earlier observation of the group that intestinal ATP levels affects viability of T_H and by this the secreted IgA. The present manuscript reports that the intestinal microbiome is responsible for ATP release and by this modulates the IgA levels in the intestinal lumen. A vaccination strategy was tested that made use of apyrase expressing attenuated *Salmonella*, resulting in decreased ATP release. Vaccination with this strain led to increased *Salmonella* specific IgA titers and protection against challenge infections.

There are several technical and conceptual issues that will need further attention.

We thank the reviewer for considering our manuscript of interest and highlighting the relevance of our findings for eliciting protective mucosal responses against enteric infections.

Specific comments:

Formally, the statement that the released ATP is bacterial origin is not sufficiently supported. Since archaeal and eukaryotic microbes are as well members of the intestinal microbiome of SPF mice, the ATP detected in ileal lumen (Fig. 1) could be as well of archaeal, fungal or protozoan origin. The authors only follow release up release by bacteria but not by other microbes in Fig. 1C.

The reviewer correctly pointed out that ATP in the ileal lumen can derive from archaea, fungi or protozoa. In fact, we did not rule out this possibility. Therefore, to address this point we performed PCR with primers specific for the three different groups of microorganisms. We were not able to amplify any DNAs for protozoa. We detected a positive signal for fungi. However, the DNA of *Saccharomyces cerevisiae* we detected was also present in the sterile diet used for feeding mice in our animal facility. Indeed, standard chow diet is enriched with yeast extract. Moreover, efforts to culture fungi did not succeed in isolating any microorganism, indicating that our spf mice are not colonized by fungi. On the contrary, we could amplify DNA of archaeal origin.

We added a sentence in the revised ms to clarify this point (page 3):

“We cannot exclude that fungi, archaea and protozoa contribute to the eATP present in the intestinal lumen. However, PCRs with primers specific for these microorganisms (ref. 7-9) provided evidence of colonization only for Archaea in our SPF mice (data not shown).”

How can the effect of purified recombinant apyrase in contrast to apyrase expressed by attenuated bacteria be explained? Apyrase expressed by recombinant bacteria will act in the periplasm and thus will result in cell-autonomous reduction of ATP release by individual bacteria. In contrast purified apyrase will act on the overall level eATP, without an effect on individual bacteria. Did the authors check if the overexpression of apyrase in *E. coli* and *S. Typhimurium* leads to release of apyrase into the culture supernatant or intestinal lumen?

We thank the referee for pointing out this issue. In fact, we consider the effect of recombinant apyrase analogous to apyrase expressed by attenuated bacteria as shown by experiments in which administration of recombinant apyrase together with PA-killed bacteria resulted in specific IgA responses analogous to immunization with live attenuated vaccines expressing apyrase (Fig. 4d and Fig. 6d). To answer to the referee concern we tested apyrase activity in the culture supernatant of the different transformants. We observed massive ATP degradation by culture medium of *E. coli*^{pApyr} as compared to *E. coli*^{pBAD28} (Fig. 2b). Apyrase activity was also detected in a crude outer membrane vesicles (OMVs) preparation from *E. coli*^{pApyr} but not *E. coli*^{pBAD28} (Fig. 2c). Moreover, the analysis of mice at 24 h after the last immunization with *E. coli* transformants (day 22 in Supplementary Fig. 3b) revealed significantly enhanced ATP degradation by ileal washes of mice immunized with *E. coli*^{pApyr} with respect to *E. coli*^{pBAD28} (figure below for referee's perusal). These data show that apyrase is released in the culture supernatant and intestinal lumen by *E. coli*^{pApyr}.

Apyrase release in the intestinal lumen. Apyrase activity expressed as percentage of input ATP in ileal washes from non-immunized mice (CTRL) and mice immunized with the indicated *E. coli* transformants at day 22 (see Supplementary Fig. 3b)

Fig. 2: the models for ATP release in panel A and B are over-simplified. There is no way ATP will directly cross inner and outer membrane. For outer membrane passage, one might consider porins. However, inner membrane to periplasm passage of ATP is very unlikely, apart from situation of membrane damage or lysis of cells.

The referee here raises an important point. Indeed, the model in figure 2 is an oversimplification because of the unlikely possibility that ATP permeate the inner membrane. However, we measured robust increases in eATP during the exponential phase of bacterial growth without concomitant increase in relative cell death (see figure below), suggesting that ATP release is associated with bacterial expansion to a variable extent depending on the particular species (Fig. 1c). In contrast to *E.coli*^{pBAD28} cultures, where the increase in eATP is readily detected, eATP is undetectable in *E.coli*^{pApyr} cultures indicating efficient degradation by apyrase (Fig. 2d and figure below for referee's perusal). According to the referee's suggestion, to be consistent with the actual knowledge and the demonstration of apyrase release (see previous point) we modified the model in figure 2; we now show the release of apyrase generating extracellular ATP and AMP in *E.coli*^{pBAD28} and *E.coli*^{pApyr} cultures, respectively. We believe our observations constitute an important starting point for future works to more thoroughly investigate ATP release from bacteria.

ATP release by *E. coli* transformants in culture and cell viability. ATP concentration (red bars) and % viable bacteria over bacterial growth (OD₆₀₀) in *E. coli*^{pBAD28} (left panel) and *E. coli*^{pApyr} (right panel) cultures.

The authors show that application of purified apyrase or apyrase producing bacteria results in increased IgA titers. Can this effect be counteracted by direct application of ATP?

To answer this point we administered a non-hydrolysable analog of ATP (ATP_γS) to WT and *P2rx7*^{-/-} mice. As shown in the figure below, we observed a significant decrease in T follicular helper (T_{fh}) cells in the Peyer's patches (PPs) of WT but not *P2rx7*^{-/-} mice following this treatment, indicating that direct application of ATP affects T_{fh} cells abundance via P2X7.

T_{fh} cells reduction by ATP_γS administration. Fold decrease of T_{fh} cells in PPs from WT and *P2rx7*^{-/-} mice treated daily with 1.25 mg ATP_γS or PBS for 14 d by oral gavage (mean±SEM, n=5; two-tailed Mann-Whitney U test, **p < 0.01).

Fig 3E: Either the scale bars are incorrect, or STM pBAD28 is twice the size as STM pApyr. It seems that the detail sections are incorrectly selected. What defines planktonic and clump? My interpretation of this experiment is that lower amounts of bacteria are present in the lumen of STM pBAD28 vaccinated mice.

We apologize for the imprecision in figure 3e (now Fig. 4e). We corrected the inset by using the correct magnification.

Individual bacteria were scored visually as planktonic, whereas aggregates of equal or more than 3 bacteria were scored as clumps. We detailed this point in the methods section (page 18). Mice received 10^7 CFUs of a 1:1 mix of fluorescent *Salmonella* and cecal content was analysed 8 hours later. The increase in clumps results in the effective increase of the amount of bacteria detected in selected fields because of aggregation, as correctly noticed by the referee. Figure 4f shows that mice immunized with S.Tm pBAD28 are characterized by higher amount of bacteria in the cecal crypts.

Fig. 3F: What defines the difference between STM pBAD28 and STM pApyr? The distribution appears rather similar. The right panel remains cryptic without further explanation of the approach of analysis. Use additional Suppl. Fig. to explain what was quantified here.

We thank the referee for remarking the lack of methodological explanation for figure 3f (now Fig. 4f) that escaped to our attention. As requested, we added Supplementary Fig. 10 to detail how total fluorescence/area was quantified in the imaged crypts.

The way the CFU counts are presented in Fig 4 B, E, H and Suppl. Fig 2 and 3 makes it difficult to estimate the effect of the vaccination regimes. It is more appropriate to display the CFU counts with log scale Y-axes. The data shown to not justify statements such as '...generated enhanced protection' since difference between mock and apyrase or vector and pApyr are rather minute.

We modified the figures as indicated by the referee. In fact, the log scale in the y axes allows a better estimation of the effect of the different immunogens in various organs.

Minor comments:

The authors used flow cytometry, not FACS in their experiments.

We modified the text.

Suppl. Fig. 1e: is total CFU/g feces, total amount of E. coli, CFU of plasmid harbouring E. coli given here?

We modified the figure legend and now state that fecal samples were plated in LB agar supplemented with 30 $\mu\text{g/ml}$ chloramphenicol and 100 $\mu\text{g/ml}$ ampicillin in order to select plasmid-harboring *E. coli*.

For Fig 2A and B, please indicate the time point of induction of Apy in the growth experiments.

We added in the figure legend that arabinose was added at time 0.

Reviewer #2 (Remarks to the Author):

Thank you for the opportunity to review this interesting and generally well-written paper.

The authors have consistently shown a link between the ectopic (i.e. introduced non-pathogen or pathogen) expression of ATP-diphosphohydrolase which reduces extracellular ATP levels, and a significant biological phenotype.

These studies were conducted to prove a novel and attractive hypothesis that it is microbiome-released ATP that drives reductions in IgA levels, to preserve the microbiome. The extension of this hypothesis is that vaccination impact can be enhanced if the microbiome produced eATP is reduced.

The evidence that there is biologically-relevant levels of microbiome-produced eATP is relatively weak and there are other explanations for the eATP observed. On balance they show that depletion of eATP increases IgA but the attempts to show it is this increased IgA that is responsible for a biological/vaccination effect on infection (e.g. 3E Clump data, the responses of 5 mice were bimodal) are less convincing. The phenomenon described may well be mediated through P2rx7, if they occur.

We thank the reviewer for considering our study interesting and the proposed hypothesis of eATP based control of adaptive mucosal immunity as novel and attractive. As shown in our previous publication (Proietti et al. Immunity 2014), P2X7 plays a central role in regulating Tfh cells abundance. From this starting point, here we addressed in depth the function of microbiota derived ATP in regulating T cell dependent mucosal immunity. Multiple evidences point to the microbiota as the origin of eATP in the small intestine: i) poor eATP in the intestinal lumen of germ-free mice (Fig. 1a); ii) acute increase of endoluminal and portal vein eATP by administration of bactericidal antibiotics (Fig. 1f); iii) reduction of eATP after prolonged administration of bactericidal antibiotics (see graph below); iv) virtual absence of ileal ATP in germ-free mice monocolonized with *E. coli*^{pApyr} but not *E. coli*^{pBAD28} (Figure 4J in Perruzza et al., Cell Reports 2017). These experiments have also shown the increase of specific SIgA in eATP-depleted mice monocolonised with *E. coli*^{pApyr} as compared to eATP-repleted mice by *E. coli*^{pBAD28} (Figure 4L in Perruzza et al., Cell Reports 2017). To definitely demonstrate that increased SIgA by depletion of eATP is responsible for the enhanced protection following vaccination, in the revised manuscript we added results obtained by vaccinating *J_H*^{-/-} mice (which lack SIgA) with *E. coli*^{pBAD28} or *E. coli*^{pApyr}. These experiments show the lack of differences in the biological effect of the two transformants in the absence of SIgA (Supplementary fig. 6f-l).

Specific issues

On a general note, the choice of axis units on the graphs seems to vary within experiments and between timepoints, making comparisons more complicated than necessary (e.g. Supp Figure 2E). I would also suggest that the Methods and Legends carry too little detail - in some cases to interpret the data, in others to allow reproduction of the data.

We thank the referee for noticing this inconsistency in the choice of axis units; we modified the figures to allow easier comparisons. Moreover, we added a number of details in the Methods section and Figure legends (in red font in the revised version) to better interpret the data and allow the reproduction of the experiments.

The source of ATP in the natural putative phenomenon of eATP regulation of IgA levels needs further exploration. There are gross differences between the GI tracts from normal animals, and those from animals lacking a microbiome. While it is possible that the ATP might come directly from bacteria, it is also possible the host homeostatic mechanisms used to control the inflammation that might be driven by the microbiome, also generate extracellular ATP. Is there a means of

removing the microbiome over an extended period, using the antibiotic cocktail described, i.e. from an adult mouse, and correlating ATP reductions with loss of this microbiome, and an increase in IgA?

The referee correctly pointed out that host homeostatic mechanisms to control possible inflammation driven by the microbiome might also generate eATP. To rule out the contribution of host cells to the endoluminal ATP measured in our assays, we administered peracetic acid killed bacteria in germ-free mice to mimic a host response to bacterial products. However, the administration of killed bacteria did not result in increase in ileal ATP, thus demonstrating that host minimally if at all contributes to intestinal ATP levels (Supplementary Fig. 8b). Moreover, in this setting, in which eATP is not detectable, immunization with both *S.Tm* pBAD28 and *S.Tm* pApyr generate similar *Salmonella* specific IgA response (Figure 6g). Prolonged VAM administration results in significant reduction of luminal ATP (see above). We have previously shown that this treatment corresponds to a significant decrease of Tfh cell in the PPs (i.e. T cell dependent IgA) because of decreased antigen burden (Perruzza et al., Cell Reports 2017, Supplementary figure 5B). These experiments indicate that antigenic stimulation is necessary to unravel the function of eATP in limiting specific IgA response. To further address the bacterial origin of eATP, we performed additional experiments that are described in the revised version of the ms (see following text):

“To further address the role of ATP released by bacteria in modulating the SIgA response, we monitored endoluminal ATP after orogastric administration of *E. coli*^{pBAD28} and *E. coli*^{pApyr} in mice maintained with Chloramphenicol and Ampicillin (CA) (a bactericidal mix active on endogenous flora but not on CA-resistant *E. coli*^{pBAD28} and *E. coli*^{pApyr}) or Penicillin/Streptomycin/Vancomycin (PSV) (bactericidal on both endogenous flora as well as *E. coli* transformants) in drinking water (Fig. 3d). Oral gavaging with *E. coli*^{pBAD28} in mice maintained in PSV as compared to CA resulted in a significant acute increase of endoluminal ATP because of bacterial lysis (Fig. 3e). Notably, the analysis of anti-*E. coli* IgA after multiple gavaging in this setting showed that the increase in eATP concomitant to *E. coli*^{pBAD28} gavaging in the presence of PSV correlated with reduced anti-*E. coli* IgA with respect to the group treated with non-bactericidal CA (Fig. 3f). In contrast, in mice colonized with *E. coli*^{pApyr}, ATP degradation by apyrase in both treatment groups (Fig. 3e) resulted in undistinguishable anti-*E.coli* IgA response (Fig. 3f). These data further show that an increased release of ATP by bacteria corresponds to a reduced generation of specific IgA.”

Could the changes (Figure 1F) generated at 3h post VAM treatment have been generated by ATP released from host cells responding the increase in PAMPs released from the bacteria killed by the antibiotics, i.e. instead or, or possibly as well as, the ATP derived from bacterial lysis?

The panel with *E. coli*^{pBAD28} in figure 3e of the revised ms shows that the release of ATP from host cells stimulated by PAMPs does not contribute to the increase in endoluminal ATP because the increase in endoluminal ATP is detectable only in the presence of bacterial lysis (administration of PSV).

The authors suggest that a reduction in eATP very quickly converts into enhanced specific IgA production. How quickly is this phenomenon occurring and would there be sufficient time to fully engage the bacterium-specific B cells, Thf etc. to drive an IgA response, after eATP is reduced through ectopic expression? It is not clear when (after immunization) these responses (e.g. Figures 2C and 2D) were measured. In situ, the relative timing of eATP regulation and microbiome-specific IgA 'suppression' by eATP is a key issue. Hypothetically, does it occur early, or with adapted microbiomes only?

We apologize for not providing enough details for the experiments we used to induce a T cell dependent IgA response against *E. coli*. We now show as Supplementary Figure 3b the diagram of the immunization protocol with *E.coli* transformants. Since *E. coli* immunization generates low titer of T cell dependent IgA after immunization (ref. 17), we immunized C57Bl/6 mice twice a week for 3 weeks and analysed Tfh cells representation (Fig. 3a) and IgA titers (Fig. 3c) as detailed in Supplementary Fig. 3b. Both these values were significantly reduced in mice immunized with *E. coli*^{pBAD28} versus *E. coli*^{pApyr}, suggesting eATP was affecting

these parameters. To address the referee’s concern about the timing of eATP regulation by apyrase bearing bacteria, we measured ileal ATP concentration 24 h after the last immunization (day 22). Endoluminal ATP was still significantly reduced in mice immunized with *E. coli*^{pApyr} versus *E. coli*^{pBAD28} (see figure), thus suggesting that enhanced IgA production is the result of protracted reduction of endoluminal ATP.

Protracted reduction of ileal eATP by *E.coli*^{pApyr}. ATP concentration in the ileum of mice immunized with *E.coli*^{pBAD28} or *E.coli*^{pApyr} at day 22 of immunization.

The *Salmonella* immunization experiments are somewhat hard to interpret. The ability to understand the nature of the experiments is somewhat hampered by the lack of description of the strains - by searching, it is clear that the ATCC 53648 is a *Salmonella enterica* var *Typhimurium* known as Chi4064, an attenuated *cya/crp* deletion mutant of *S. Typhimurium* SR-11, developed ? by Roy Curtiss. At 28 days after 3 vaccinations, 3 days apart, the Chi4064-vaccinated mice were challenged with ? standard *S. Typhimurium* SL1344, a more common strain of *S. Typhimurium*. Many previous studies would suggest that mice vaccinated for 1 month with attenuated *Salmonella* mutants like Chi4064 would have had vaccination-derived bacteria still in the tissues, making it more difficult to identify vaccine- and challenge-derived bacteria when estimating challenge effects.

We apologize for lack of precision in defining the various bacterial strains used in this study. We provide Supplementary Table 1 with all the requested details. To discriminate between virulent *Salmonella* SL1344 used for the challenge (Streptomycin resistant) and attenuated strain used for immunization (Chloramphenicol/ampicillin resistant), the analysis of CFUs in tissues was performed in MacConkey agar plates containing 50 µg/ml Streptomycin to select virulent bacteria (detailed in Methods section).

The authors conclude that the immunity observed is due to IgA, but it could be to other immune responses such as provided by CD4+ T cells; the *Rag1* mutation might equally impact these other cell types. It cannot safely be assumed that the apparent reductions in bacterial counts in the immunised mice are driven by differentially heightened IgA responses, absent some more targeted analysis.

We thank the referee for raising this important point and allowing us to better address the role of apyrase-induced IgA in protecting from enteropathogenic infection. We immunized $J_H^{-/-}$ mice, which lack Ig, with attenuated *S. Tm*^{pBAD28} and *S. Tm*^{pApyr} and then challenged them with virulent *S. Tm*. As shown in Supplementary figure 6 f-l, both non-immunized and immunized mice were equally infected by virulent *S. Tm*, showing that SIgA is responsible for protection upon oral immunization. Lack of IgA renders mice equally susceptible to infection irrespective of immunization with control or apyrase bearing bacteria. Moreover, the analysis of effector/memory T cells as well as IL-17, IFN-γ and TNF-α secreting T cells did not reveal any difference between mice immunized with *S. Tm*^{pBAD28} and *S. Tm*^{pApyr} (data not shown).

The use of mCherry and GFP-tagged bacteria and photon microscopy is an interesting means of dissecting the immune response, but the genotype of the 'att' strain is cryptic, and the description of this experiment is too limited to understand the experimental design and interpretation.

We provide Supplementary Table 1 (see above), in which also abbreviations for the strains used in the study are specified. We added Supplementary Fig. 10 to explain how total fluorescence/area was quantified in the imaged crypts.

From a technical perspective there are some minor issues.

1. The use of an arabinose-dependent promoter for ectopic expression of the *phoN2* means that transcription of *phoN2* quickly ceases when arabinose is withdrawn and bacterial replication occurs - was arabinose fed to the infected mice?

We mention in the methods section that 0.3% arabinose was added in the drinking water during immunization (page 17).

2. Is the luminal eATP released from the microbiome freely basement- and cell-membrane permeable - can it reach the tissues where IgA is to be produced - lumen-administered labelled ATP might reveal this.

To address this issue, as suggested by the referee, we gavaged mice with Mant-ATP triethylammonium and evaluated intestinal permeability by quantifying fluorescence intensity in serum from portal vein. The figure below shows the kinetics of Mant-ATP fluorescence, which indicates that ATP permeates the intestinal epithelium.

Intestinal permeability to Mant-ATP. At the indicated time after orogastric administration of 10µM Mant-ATP, C57BL/6 mice were sacrificed and serum from portal vein was collected and tested to quantify the Mant-ATP absorption. The samples were excited at 355nm and fluorescence signal was collected at 448 nm.

3. Is the enhanced level of IgA a result of increased IgA transport via the *plgR*, rather than synthesis?

Figure 4c provides a quantitative analysis of plasma cells specific for Salmonella LPS, suggesting that the increase in Salmonella specific IgA detected in mice immunized with apyrase bearing bacteria is due to expansion of IgA secreting cells.

4. The bacterial species selected for 'specificity checking' after *E. coli* immunisation include genera where significant *E. coli* cross reactions might be expected because of genome similarity, such as *Salmonella* and *Klebsiella*, especially if a K-12 lab variant was used (these are usually LPS-deficient) - are the elevated anti-bacterial responses displayed in 2C and 2D LPS specific? This is easily tested. Lastly, the species selected for comparison (2E) do not necessarily form part of the murine microbiome; were the specific isolates tested derived from their animal house and was a microbiome analysis conducted to ensure that the strains tested were relevant?

We apologize for our lack of precision. The *E. coli* strain used in this study is DH10B, which is characterized by "normal" LPS composition (Durfee et al., J Bacteriol 190: 2597-2606, 2008). Therefore, the secretory IgA generated by immunizing with DH10B transformants could be also elicited by LPS and was not cross-reactive with *Salmonella* (previous Fig. 2e, now Supplementary Fig. 4a). As correctly pointed out by the referee the species selected for comparison (previous Fig. 2e, now Supplementary Fig. 4a) do not necessarily form part of the murine microbiome. To address the IgA response to the microbiome, we tested the binding of SIgA present in ileal wash from immunized mice on microbiota colonizing our spf mice. As shown in Supplementary figure 4b, we did not detect any difference in SIgA binding to the endogenous flora with intestinal washes from either non-immunized mice or mice immunized with different *E. coli* transformants. This experiment shows that the immunization protocol does not enhance the secretory IgA against the microbiota but selectively target the immunogen.

Reviewer #3 (Remarks to the Author):

This manuscript builds on previous work from the same group, where they showed that extracellular eATP stimulation via the P2RX7 receptor lead to apoptotic death of Tfh cells in the Peyer's patches. In the present study, they investigate whether eATP from intestinal microbiota might therefore inhibit specific IgA responses towards intestinal bacteria. They report that ileal bacteria produce eATP, some of which reaches the local circulation, and that treatment with antibiotics increases eATP release by intestinal bacteria, resulting in increased Tfh death and reduced IgA responses. Conversely, expression of apyrase (an ATP-degrading enzyme) in E. coli led to improved induction of specific IgA antibodies.

To demonstrate the potential utility of their findings, they turn to a Salmonella (S.Tm) model, and report that vaccination with an attenuated S.Tm strain expressing apyrase again resulted in increased production of high affinity secretory IgA that confers enhanced protection from challenge with virulent S.Tm. They also show that administration of apyrase extract during vaccination with inactivated S.Tm leads to enhanced protective IgA responses. This study reveals a novel pathway in which protective secretory IgA responses are limited by eATP produced by intestinal bacteria and shows that blockade of this pathway by apyrase can improve the response to oral vaccination.

Overall, this is a very interesting and high-quality study that is clearly presented, well controlled and with conclusions that are strongly supported by the data.

I have a few specific comments that the authors could consider to further clarify and/or strengthen their study.

1. The data in Fig. 1D-G show that treatment with VAM leads to increased eATP release from bacteria both in vitro and in vivo. The in vivo increases are less pronounced and the ileal and portal vein ATP levels in the VAM-treated group (Fig. 1F) look very comparable to those found in the untreated SPF mice in Fig. 1A,B. These observations suggest that there was some intra-experimental variability in the ATP levels. This brings up one noticeable weakness in the manuscript, namely that the reproducibility of the findings and the number of independent experimental repeats were not stated. The figures appear to show representative experiments and often have quite low n values. Therefore, all the figure legends should include full information on how many times each finding was confirmed and whether the results shown are representative, or instead represent pooled values from multiple independent experiments.

We thank the reviewer for the careful analysis of our data. In fact, eATP concentrations in untreated SPF mice (Fig. 1a) and after VAM treatment (Fig. 1f) look comparable. As correctly hypothesized by the referee, we observed inter-experimental variability likely because of some variation in microbiota and susceptibility of ATP to some hydrolysis during organ manipulation. The experiments mentioned by the referee were performed two years apart and this might explain the observed variability. In fact, we decided to show single representative experiments (with n=5 in this case) instead of cumulated fold increase of ATP concentration from multiple experiments. We apologize for the lack of details on experimental repeats. We now included this information in the figure legends.

2. The acute exacerbation of intestinal ATP levels following VAM treatment was accompanied by increased Annexin V expression by Tfh cells (Fig. 1F,G). The authors suggest that, "... antibiotic treatment may negatively effect the induction of T-cell dependent intestinal immunity in these infections". However, is it not likely that VAM treatment leads to an acute increase in intestinal ATP, followed by a subsequent decrease in intestinal ATP (due to depletion of microbiota)? Therefore, with prolonged VAM treatment (more representative of a clinical scenario), one might expect to see a rebound in Tfh cells as the ATP levels decrease. Therefore, analogous to the excellent responses of GF mice to vaccination (Fig. 4G,H), one might expect mice treated with VAM for a longer period to make enhanced responses to oral vaccination?

The referee correctly hypothesized that VAM treatment lead to an acute increase in intestinal ATP, followed by a subsequent decrease in intestinal ATP due to depletion of microbiota. Prolonged VAM administration results in significant reduction of luminal ATP (see figure). Concomitantly, we have previously shown that this treatment corresponds to a significant decrease of Tfh cell in the PPs (i.e. T cell dependent IgA) because of decreased antigen burden (Perruzza et al., Cell Reports 2017, Supplementary figure 5B). These experiments indicate that antigenic stimulation is necessary to unravel the function of eATP in limiting specific IgA response.

Microbial origin of ileal ATP. Fold decrease in ileal ATP concentrations in SPF mice after 15 days of daily orogastric gavage with PBS or VAM.

To further address the role of bacterial eATP in modulating SIgA response, we performed additional experiments that are described in the revised version of the ms (see following text):

“To further address the role of ATP released by bacteria in modulating the SIgA response, we monitored endoluminal ATP after orogastric administration of *E. coli*^{pBAD28} and *E. coli*^{pApyr} in mice maintained with Chloramphenicol and Ampicillin (CA) (a bactericidal mix active on endogenous flora but not on CA-resistant *E. coli*^{pBAD28} and *E. coli*^{pApyr}) or Penicillin/Streptomycin/Vancomycin (PSV) (bactericidal on both endogenous flora as well as *E. coli* transformants) in drinking water (Fig. 3d). Oral gavaging with *E. coli*^{pBAD28} in mice maintained in PSV as compared to CA resulted in a significant acute increase of endoluminal ATP because of bacterial lysis (Fig. 3e). Notably, the analysis of anti-*E. coli* IgA after multiple gavaging in this setting showed that the increase in eATP concomitant to *E. coli*^{pBAD28} gavaging in the presence of PSV correlated with reduced anti-*E. coli* IgA with respect to the group treated with non-bactericidal CA (Fig. 3f). In contrast, in mice colonized with *E. coli*^{pApyr}, ATP degradation by apyrase in both treatment groups (Fig. 3e) resulted in undistinguishable anti-*E. coli* IgA response (Fig. 3f). These data further show that an increased release of ATP by bacteria corresponds to a reduced generation of specific IgA.”

Finally, in Fig. 1G it would be good to include some representative FACS plots of the Tfh staining, as the magnitude of the increase seems quite modest.

As the referee surely observed, there is substantial variability between samples that likely derives from the experimental procedure, in which the acute antibiotic treatment could affect Tfh cells response with slightly different efficacy in distinct mice. Nevertheless, the statistical analysis (two-tailed Mann-Whitney U-test) revealed significant difference between PBS and VAM treated animals. We revised figure 1g to provide representative analyses in flow cytometry of Annexin V staining. In addition, we show the staining used to gate Tfh cells as Supplementary Fig. 1.

3. The data in Fig. 2 show that *E. coli* pApyr induces higher IgA responses than *E. coli* pBAD28. Does this correlate with decreased ileal and portal vein ATP concentrations?

We provide the figure below for referee’s perusal showing decreased ileal ATP concentration at 24 h after the last gavage (day 22) with *E. coli*^{pApyr} versus *E. coli*^{pBAD28} (panel a), thus correlating with higher IgA response. We were not able to detect the same significant reduction in the portal vein blood probably because the contribution of absorbed ATP originating from endogenous bacteria does not allow appreciating any difference at this site. However, the same analysis in a germ free mice monocolonized with *E. coli* transformants (28 days after single gavage colonization) revealed the significant reduction of ileal ATP (Figure 4J in Perruzza et al., Cell reports 2017) together with concomitant decrease of portal vein ATP in animals colonized with *E. coli*^{pApyr} compared to *E. coli*^{pBAD28} (panel b in the figure below).

Decreased ileal and portal vein eATP by *E. coli*^{pApyr} (a) ATP concentration in the ileum of SPF mice immunized with *E. coli*^{pBAD28} or *E. coli*^{pApyr} at day 22; (b) ATP concentration in portal vein of GF mice after 28 days of monocolonization with *E. coli*^{pBAD28} or *E. coli*^{pApyr}.

What about Tfh levels and Annexin V staining in the PP? It would be good to include these data - they could replace Fig. 2E which could easily be moved to the Supplemental data.

The analysis of Tfh cells and Annexin V in Tfh cells at 24 h after the last gavage (day 22) showed the significant increase of Tfh cells and significant decrease of Annexin V staining in Tfh cells in mice immunized with *E. coli*^{pApyr} versus *E. coli*^{pBAD28}. As suggested by the referee, we include these data as figure 3a and moved analysis of bacteria in flow cytometry (previous Fig. 2e) to supplemental data (Supplementary Fig. 4a). We thank the referee for this suggestion, which improved our manuscript.

Did you ever try co-infection with both strains of *E. coli*? Presumably the presence of the *E. coli* pApyr strain would overcome any inhibitory effects of the *E. coli* pBAD28 strain?

This is an interesting point that we addressed in figure 6, where we show that addition of apyrase from *E. coli* p^{Apyr} is sufficient to confer the same immunogenicity as live *S. Tm* p^{Apyr} to peracetic acid killed *S. Tm* p^{BAD28}, thereby indicating that apyrase is responsible for the enhanced SIgA generation and protection.

4. Even having carefully read the methods, it is difficult to precisely understand why the IgA antibody titers are expressed as “ml/ng”. I assume that the “ng” refers to the total IgA concentration, but what does the “ml” refer to?

We apologize for this inconsistency. We now provide more details on the method used to quantify the IgA antibody titer and include the reference (ref. 36), which describes this analysis in depth.

Methods section (page 15-16):

“The concentration of total antibody titer required to achieve a given MFI (for example =500) was calculated by re-arrangement of the fitted 4-parameter logistic equation for each samples. As this value is low where a strong antibody response is present, the inverse of this value was plotted. Thus titers are calculated as the inverse total antibody concentration required to achieve a given MFI. The y-axis value chosen as “above background” necessarily varies between experiments due to the flow cytometer settings, but is constant within any one analysis.”

5. I would recommend changing the scale on the histological data shown in Supplemental Fig.3A, as it makes it look like there is a trend to reduced pathology in the *S. Tm* pApyr vaccinated group. Having the same scale as used in the histology graph presented in Supplemental Fig.2C would emphasize the similarities across the groups.

We are grateful to the referee for identifying this mistake in the setting of the scale in Supplementary Fig. 3a (now Supplementary Fig. 6a). We have modified the graph to adopt a similar scale used in Supplementary Fig. 2c (in the revised ms Fig. 5a).

6. In Supplemental Fig.3.F-I the presented data indicate that P2RX7^{-/-} mice are equally protected by *S. Tm* pBAD28 and *S. Tm* pApyr vaccination. However, in Fig.S3F, it looks like they make more IgA after pApyr immunization. Does this suggest some P2RX7-independent effects of eATP on SIgA responses? If so, this should be discussed by the authors.

As correctly observed by the referee, P2rx7^{-/-} mice vaccinated with *S. Tm* p^{Apyr} show a slight increase, albeit not significant, of IgA specific for *Salmonella* with respect to mice immunized with *S. Tm* p^{BAD28}. At the moment, we do not have an explanation for this phenomenon. However, we did not observe any difference in the protection from *Salmonella* infection between the two groups of immunized mice.

7. The experiments in Fig.4D-F show that addition of apyrase extract results in enhanced protection following vaccination with fixed *S. Tm*. This suggests that apyrase could be utilized as an adjuvant for oral vaccination. This raises the issue as to whether apyrase is immunogenic itself and are specific IgA antibodies produced against apyrase?

We thank the referee for raising this important point. We tested small intestinal lavages for the presence of anti-apyrase IgA in ELISA on purified apyrase from *E. coli* p^{Apyr}. This analysis did not provide any evidence that our immunization protocol elicited a specific secretory IgA response against apyrase. We provide these results for referee’s perusal below and added this information as data not shown in the revised ms (page 9).

Immunization with apyrase extract does not elicit anti-apyrase IgA. Determination of anti-apyrase IgA in small intestine’s washes of non-immunized mice (CTRL) or mice immunized with PA-*S. Tm* p^{BAD28} together with crudely-purified apyrase (APY extract) or mock extract (mock extract). Samples were tested in ELISA on plates coated with purified apyrase. An anti-apyrase control serum was used as positive control.

Reviewers' comments:

Reviewer #1 (Remarks to the Author):

Thank you for addressing the concerns raised by additional experiments and revision of the manuscript.

Reviewer #2 (Remarks to the Author):

Source of ATP

The authors have attempted to show that the eATP that appears to drive their phenomenon comes from the microbiome. These efforts include reducing the microbiome by antibiotic treatment and studies in Germ Free mice. However, none of the experiments would exclude the possibility that, in response to microbiome colonisation, the colonised epithelia produce ATP which then acts, as hypothesized by the authors, as a regulator of IgA.

If the ATP is epithelia-derived, and is produced as a consequence of microbiome colonisation, the levels would rise in SPF animals, and decrease in concert with reductions in levels of the bacterial microbiome, e.g. through antibiotic treatment.

Figure 1a (GF) shows a small but measurable degree of ATP production in Germ Free mice, presumably from the host.

It is well documented that there is rapid turnover of the gut epithelia and this turnover, potentially releasing ATP, is increased by the presence of the microbiome, possibly through lactate production (Okada, Nature Comms 4: Article 1654 2013).

The observation that bacteria produce extracellular ATP *ex vivo* in culture is only indirect evidence that it might happen *in vivo*. As raised already, ATP generated from live (especially Gram negative) bacteria would need to cross multiple membranes to effect a response; this would be less of a challenge if the ATP came from the host side.

Other issues

The Salmonella protection experiments need a clearer description and explanation. The vaccination and challenge procedures need to be more explicit – which animals were vaccinated with S.Tmatt, and whether this vaccine was modified with pApyr. The enhanced protective effect also appears to be transitory – while the modified (+ pApyr) vaccine may have induced stronger IgA responses and the challenge was restricted from the mucosal tissues like the GALT early after infection, did this enhanced performance translate to infection immunity?

Lines 151-152 imply that protection against Salmonella infections is mediated by IgA. There is limited to no evidence for this or indeed antibody-mediated protection in BL6 mice. Published data shows that protection against infection is largely mediated by CD4+ T cells and to a lesser extent NK cells (groups of McSorely, Mastroeni et al.). The animals were analysed post-challenge over the first 48 hours only. This might be a suitable window for analysis of the epithelia infection but solid systemic infection generally takes 3-4 days at least. The authors should reconcile Fig 6b and h, where the Apyr containing vaccine worked as efficiently as the pBAD (empty vector) vaccine, with Figure 5e, where the vaccine carrying Apyr induced ? better protection.

The attribution of the immune effects of immunization based simply on RAG1 and IgH KOs is problematic. The RAG1 KOs retain other cellular mechanisms that don't require TCR rearrangement e.g. NK cells, and the IgH KOs have a ? B-cell APC function which is known to be

important in immunity (Nanton et al, J Immunol 189:5503, 2012)

All of the histology in 5a in 6 and the Supps looks very similar at the magnification that is presented. The features that are arrowed are not distinguishable.

Minor issues

Axis manipulation

The authors have used adjusted axis parameters to try to highlight effects – Figures 3a (both panels), 3f, 4b (LHS), that is unnecessary. Some are odd intervals (e.g. Figure 4b, RHS), 5a box plots of 'histo scores' (3 units?). The vertical axis should start at 0 or 1, depending on whether they are arithmetic or logarithmic, respectively.

S.Tmatt description needs to be added to the Bacterial Strains methods section.

Some of the mice used seem to have a B129 genetic contribution. This can confound interpretation. What was the Nramp analysed in these mice?

Reviewer #1:

Thank you for addressing the concerns raised by additional experiments and revision of the manuscript.

We thank the reviewer for the previous valuable comments.

Reviewer #2:

Source of ATP

The authors have attempted to show that the eATP that appears to drive their phenomenon comes from the microbiome. These efforts include reducing the microbiome by antibiotic treatment and studies in Germ Free mice. However, none of the experiments would exclude the possibility that, in response to microbiome colonisation, the colonised epithelia produce ATP which then acts, as hypothesized by the authors, as a regulator of IgA.

If the ATP is epithelia-derived, and is produced as a consequence of microbiome colonisation, the levels would rise in SPF animals, and decrease in concert with reductions in levels of the bacterial microbiome, e.g. through antibiotic treatment.

Figure 1a (GF) shows a small but measurable degree of ATP production in Germ Free mice, presumably from the host.

It is well documented that there is rapid turnover of the gut epithelia and this turnover, potentially

releasing ATP, is increased by the presence of the microbiome, possibly through lactate production (Okada, Nature Comms 4: Article 1654 2013).

As correctly pointed out by the reviewer, a fraction of intestinal eATP could derive from the epithelium. In fact, in SPF mice after prolonged antibiotics administration (see supplementary figure 1a of the revised ms) and germ-free mice (see figure 1a), eATP can be detected at much lower albeit detectable concentrations than SPF mice. To address the Reviewer's concerns regarding the potential impact of rapid epithelial gut turnover and its effects on ATP production, we induced epithelial regeneration in the ileum by starvation and re-feeding, as described by Okada et al. (Nature Comms 4: 1654, 2013). In SPF mice, starvation induced a significant decrease in the percentage of Ki-67+ (i.e. proliferating) EpCAM+CD45- cells. Mice refeeding coincided with a significant increase in proliferating enterocytes, as described in the colon by Okada et al. However, these variations in epithelial turnover did not correspond to significant variations in the levels of ileal eATP. The same experiment performed in mice treated for 15 days with vancomycin/ampicillin/metronidazole (VAM) did not result in substantial variations in the percentage of Ki-67+ enterocytes with respect to untreated SPF mice, as previously described in germ-free (GF) mice (Stecher et al., Infection and Immunity, 73: 3228-3241, 2005, see figure 7 panels G and P). Differently from SPF mice, in the absence of bacteria, starvation did not affect Ki-67+ cells. However, the concentration of ileal ATP was dramatically reduced, thereby suggesting that the great majority of eATP measured in the ileal lumen is of bacterial origin. We describe these experiments in the Results section (page 3,4) and added Supplementary figure 1a to document these findings.

The observation that bacteria produce extracellular ATP ex vivo in culture is only indirect evidence that it might happen in vivo. As raised already, ATP generated from live (especially Gram negative) bacteria would need to cross multiple membranes to effect a response; this would be less of a challenge if the ATP came from the host side.

To show that bacteria-derived ATP in the ileum can reach tissues where IgA is to be produced, we previously performed the experiment suggested by the Reviewer and gavaged mice with Mant-ATP triethylammonium. The analysis of Mant-ATP fluorescence in serum from portal vein suggests that ATP permeates the intestinal epithelium.

Other issues

The Salmonella protection experiments need a clearer description and explanation. The vaccination and challenge procedures need to be more explicit – which animals were vaccinated with S.Tmatt, and whether this vaccine was modified with pApyr. The enhanced protective effect also appears to be transitory – while the modified (+ pApyr) vaccine may have induced stronger IgA responses and the challenge was restricted from the mucosal tissues like the GALT early after infection, did this enhanced performance translate to infection immunity?

In the Methods section (page 17), we detailed that “For vaccination with S.Tm transformants, S.Tm^{pBAD28} and S.Tm^{pApyr} were harvested by centrifugation, washed in sterile PBS and 5×10^9 CFUs administered to mice by orogastric gavage. The procedure was repeated every 3 days for 3 times. On day 28, mice were used for infection experiments.” To more explicitly show the method, we added the diagram of immunization and challenge as Supplementary Fig. 5b. We describe in each figure legend the strain of mice used in the experiment. This information is reiterated in the respective panel.

We used the ***Salmonella enterica* serovar Typhimurium (S.Tm) colitis model** induced by pre-treatment with streptomycin. To address immunity to infection we measured GALT colonization and

dissemination of virulent bacteria in the spleen and liver. Immunization with the attenuated S.Tm strain (ATCC 53648) carrying apyrase-bearing pHND10 (S.Tm^{pApyr}) induced enhanced sIgA response and protection from Salmonella infection.

Lines 151-152 imply that protection against Salmonella infections is mediated by IgA. There is limited to no evidence for this or indeed antibody-mediate protection in BL6 mice. Published data shows that protection against infection is largely mediated by CD4+ T cells and to a lesser extent NK cells (groups of McSorely, Mastroeni et al.).

Differently from models of systemic infection with Salmonella, where protection is provided by CD4+ and NK cells, the protection from infection in *Salmonella enterica* serovar Typhimurium (S.Tm) colitis model, is mainly mediated by sIgA (ref 3). In addition, *Rag-1*^{-/-} mice, in which NK cells are functioning (Supplementary figure 6 a-e) and *Igh*^{-/-} mice, in which both NK and CD4+ cells can be activated (Supplementary figure 6 f-l), were not protected by the vaccination.

The animals were analysed post-challenge over the first 48 hours only. This might be a suitable window for analysis of the epithelia infection but solid systemic infection generally takes 3-4 days at least.

In Switzerland, the Federal Veterinary Authority does not allow the protraction of *Salmonella enterica* serovar Typhimurium (S.Tm) colitis model over 48 h post infection. However, this time window allows the analysis of dissemination of the pathogen to systemic organs, including spleen and liver.

The authors should reconcile Fig 6b and h, where the Apyr containing vaccine worked as efficiently as the pBAD (empty vector) vaccine, with Figure 5e, where the vaccine carrying Apyr induced ? better protection.

In figure 6b, we show the lack of difference in the protective effect of killed vaccines either bearing or not the apyrase gene in SPF mice. In figure 6h, we show the lack of difference in the protective effect of killed vaccines either bearing or not the apyrase gene in GF mice. The difference in the efficacy of vaccination obtained with killed bacteria versus live attenuated bacteria expressing apyrase (Figure 5e) depends on the lack of influence of killed bacteria on ileal eATP, that is not the case for apyrase expressing live bacteria, where the presence of the enzymatic activity decreases eATP levels during immunization.

The attribution of the immune effects of immunization based simply on RAG1 and IgH KOs is problematic. The RAG1 KOs retain other cellular mechanisms that don't require TCR rearrangement e.g. NK cells, and the IgH KOs have a ? B-cell APC function which is known to be important in immunity (Nanton et al, J Immunol 189:5503, 2012)

The paper by Nanton et al. addresses CD4+ and CD8+ T cell response in the spleen and protection from systemic infection by virulent Salmonella in mice immunized intravenously with attenuated Salmonella. This experimental model is different from the streptomycin model of non-typhoidal salmonellosis we used to assess the protective effect of sIgA induced by oral immunization on colitis and systemic dissemination of the pathogen.

All of the histology in 5a in 6 and the Supps looks very similar at the magnification that is presented. The features that are arrowed are not distinguishable.

We provided this magnification because it allows visualizing all the pathological aspects we used to produce the histological score: submucosal edema, neutrophils aggregates, epithelial defects and abundance of goblet cells.

Minor issues

Axis manipulation

The authors have used adjusted axis parameters to try to highlight effects – Figures 3a (both panels), 3f, 4b (LHS), that is unnecessary. Some are odd intervals (e.g. Figure 4b, RHS), 5a box plots of 'histo scores' (3 units?). The vertical axis should start at 0 or 1, depending on whether they are arithmetic or logarithmic, respectively.

In fact, the axis parameters were chosen to provide the most meaningful visualization of the described phenomenon, the significance of which is given in any case by the statistical analysis.

S.Tmatt description needs to be added to the Bacterial Strains methods section.

The S.Tmatt strains used in our study are detailed in Supplementary Table 1.

Some of the mice used seem to have a B129 genetic contribution. This can confound interpretation. What was the Nramp analysed in these mice?

In fact, we reported the original nomenclature of mice strain provided by the vendor. $P2rx7^{-/-}$ mice were backcrossed to C57BL/6 mice for 9 generations many years ago when we started breeding them at the Institute for Research in Biomedicine.

In summary, we thank the reviewer for their continued interest in exploring the epithelial versus bacterial origin of intestinal eATP: having addressed both the original and new set of their concerns, we hope the reviewer will acknowledge that we have conducted new experiments that he/she requested - all of which have reinforced our interpretation - and that remaining questions will need to await future studies. We are grateful to the reviewers and editor for their valuable input and recommendations. The new data and revised text have broadened the impact of our findings and tightened the interpretation of the work.

REVIEWERS' COMMENTS:

Reviewer #2 (Remarks to the Author):

The authors have attempted to rebut all the issues I raised in my review, and re-review.

The issue of the source of the elevated levels of eATP, while not definitively resolved, is dealt such that bacteria are the most likely source of the eATP, the effects of which drive the rest of the paper.

It is unfortunate that Swiss Ethics do not allow an analysis as to whether the effects observed have any effect on invasive disease outcomes.

The documentation is of high quality, as is the statistical analysis.